# Organisation of the orthobunyavirus tripodal spike and the structural changes induced by low pH and K$^+$ during entry

Samantha Hover [1,2], Frank W. Charlton[1,2], Jan Hellert [3], Jessica J. Swanson[1,2], Jamel Mankouri [1,2] ✉, John N. Barr [1,2] ✉ & Juan Fontana [1,2] ✉

Following endocytosis, enveloped viruses employ the changing environment of maturing endosomes as cues to promote endosomal escape, a process often mediated by viral glycoproteins. We previously showed that both high [K$^+$] and low pH promote entry of Bunyamwera virus (BUNV), the prototypical bunyavirus. Here, we use sub-tomogram averaging and AlphaFold, to generate a pseudo-atomic model of the whole BUNV glycoprotein envelope. We unambiguously locate the Gc fusion domain and its chaperone Gn within the floor domain of the spike. Furthermore, viral incubation at low pH and high [K$^+$], reminiscent of endocytic conditions, results in a dramatic rearrangement of the BUNV envelope. Structural and biochemical assays indicate that pH 6.3/K$^+$ in the absence of a target membrane elicits a fusion-capable triggered intermediate state of BUNV GPs; but the same conditions induce fusion when target membranes are present. Taken together, we provide mechanistic understanding of the requirements for bunyavirus entry.

The *Bunyavirales* constitute the largest order of enveloped negative-sense RNA viruses with over 500 named isolates. Within this group, members of the *Hantaviridae*, *Nairoviridae*, *Phenuiviridae* and *Peribunyaviridae* families possess tri-segmented genomes[1] that minimally encode an RNA-dependent RNA-polymerase (L), a nucleocapsid (N) protein that encapsidates the genomic segments forming ribonucleoproteins (vRNPs), and two viral glycoproteins (GPs) Gn and Gc, which form spikes projecting from the viral envelope[2]. Several members of these families are serious human pathogens, including La Crosse virus (LACV) (*Peribunyaviridae* family) which causes severe neurological encephalitis and Crimean-Congo haemorrhagic fever virus (*Nairoviridae* family) which carries a 30% fatality rate. However, no vaccines or antiviral therapies to prevent their associated disease in humans have been approved[3,4].

Bunyaviruses enter cells through the endocytic system, releasing their genomes following fusion between their envelope and vesicular membranes, mediated by the Gn-Gc spikes, which together comprise the fusion machinery. For the prototypic model peribunyavirus, Bunyamwera virus (BUNV), a low-resolution sub-tomogram average (STA) of the BUNV Gn-Gc (~30 Å) previously revealed an unusual tripod-like structure of the GPs, forming a local lattice-like arrangement of trimers on the virus surface[5]. Three regions of the GP tripod were termed the head (apex of the tripod; viral membrane-distal), stalk (connects head-floor) and the floor region (membrane-proximal). More recently, it was identified that the head and stalk regions are formed by the first half of the Gc ectodomain by solving the crystal structures of the head domains from the peribunyaviruses BUNV, Oropouche virus (OROV) and LACV, and the head/stalk domains from Schmallenberg virus (SBV)[6]. The membrane-proximal floor region is therefore predicted to contain the remaining half of the Gc ectodomain, which includes the fusion domain, and the entire Gn ectodomain. While there are no published structures of Gn, an AlphaFold model of LACV Gn alone was previously generated[7], however without a complete structure of Gn-Gc or a higher resolution map the interpretation of this model is limited.

[1]School of Molecular and Cellular Biology, Faculty of Biological Sciences, Leeds, United Kingdom. [2]Astbury Centre for Structural and Molecular Biology, University of Leeds, LS2 9JT Leeds, United Kingdom. [3]Centre for Structural Systems Biology, Leibniz-Institut für Virologie (LIV), Notkestraße 85, 22607 Hamburg, Germany. ✉e-mail: bms9jm@gmail.com; J.N.Barr@leeds.ac.uk; J.Fontana@leeds.ac.uk

In structures of related bunyavirus Gn-Gc complexes (*Phenuiviridae*: Rift Valley fever virus (RVFV) and *Hantaviridae*: Tula virus (TULV) and Andes virus (ANDV)) the fusogenic protein Gc adopts a class II fusion fold, and Gn acts as the shielding subunit typically overtopping Gc[8,9]. Thus Gn is likely involved in protecting the fusion loops and mediating initial contact with the target cells[10,11]. Compared to these orthologous systems, peribunyaviruses possess a relatively small Gn ( ~ half the size; 186 residue ectodomain for BUNV) and a Gc around twice the size (910 residue ectodomain for BUNV) (Fig. 1a)[10,11]. This suggests that the functions of these proteins may not be identical across bunyavirus families, which coupled with the scarcity of

peribunyavirus structures available has limited our structural understanding.

The cues that trigger fusion often result from endosomal maturation, which might involve a gradual drop in intraluminal pH, changes in concentrations of other ions, or alterations to the lipid and protein composition of endosomal membranes[12], dictating when and where fusion occurs. This prevents premature fusion in inappropriate compartments, which would jeopardize virus entry and infection[13]. For bunyaviruses, the events that lead to spike fusogenesis are incompletely characterised and indeed the structural steps in membrane fusion are often inferred from pre-fusion and post-fusion structures, and

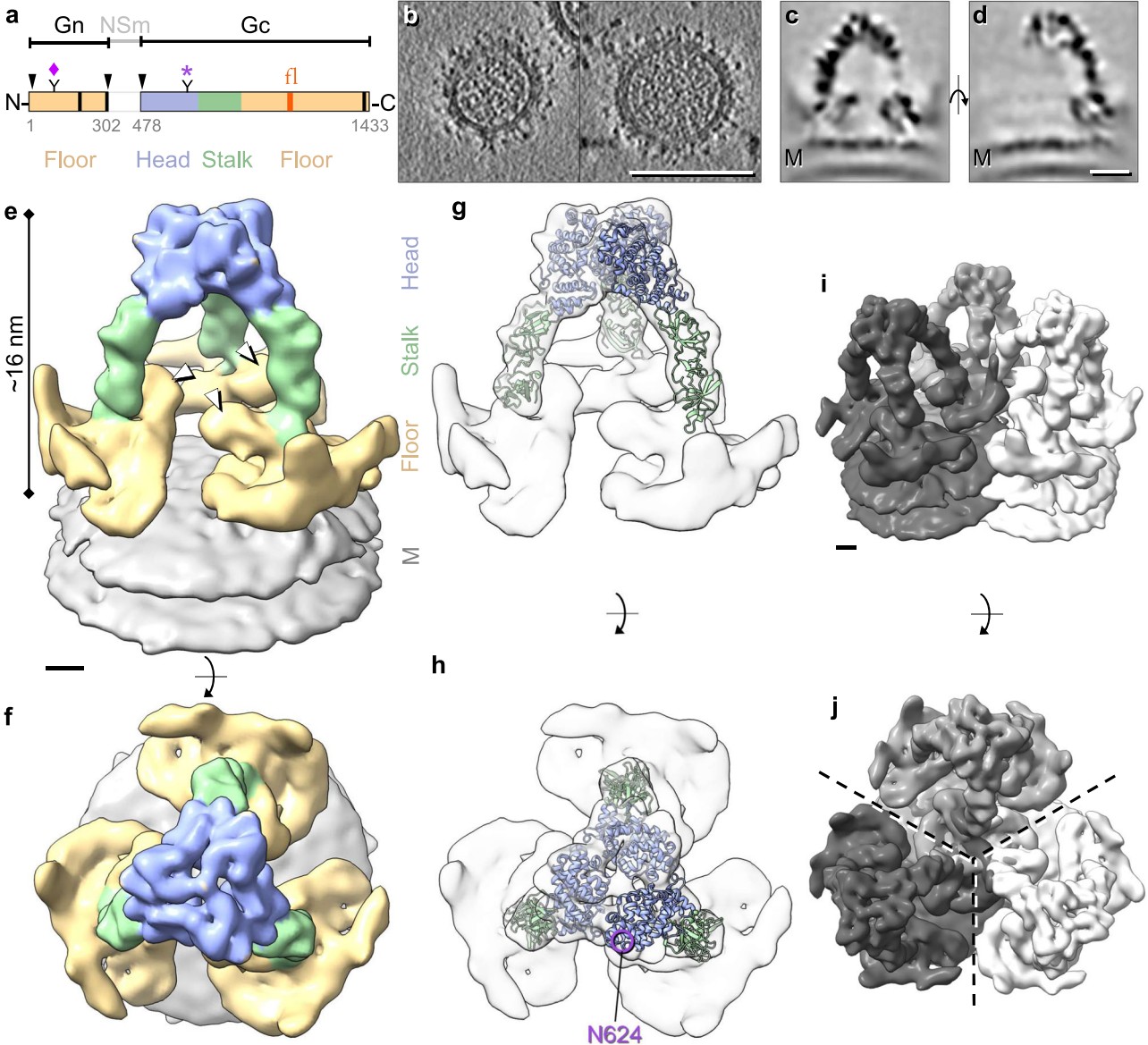

**Fig. 1 | BUNV GPs exhibit a trimeric tripod-like architecture. a** Schematic of the M segment polyprotein constituted by Gn, NSm (non-structural protein) and Gc. The Gc protein is comprised of three sections: the head (blue), stalk (green) and floor (gold) regions. The fusion loop (fl; orange) resides in the floor region. Arrowheads indicate cleavage sites, black bars the predicted TMDs, numbers the amino acid residues, and 'Y' the glycans N60 (Gn, ♦) and N624 (Gc, *)[6,31,32,34]. **b** Cryo-ET tilt series were collected of BUNV virions (buffer: pH 7.3/no K+), then 3D tomographic reconstructions were calculated (14 tomograms). Slices through tomograms of two virions are shown (representative of 71 virions, full tomogram section in Supplementary Fig. 5d). **c, d** STA of over 20,000 GP spikes aligned through iterative refinements, generating a ~ 16 Å average (GS-FSC). Panels are

sections through the electron density averages, showing a trimeric spike on top of the viral lipid bilayer (M; membrane). **e, f** Isosurface rendering of the GP trimer, identifying the three regions; head (light blue), stalk (light green) and floor (gold; where Gn also resides) on top of the viral membrane (grey). White arrowheads indicate a previously unresolved region in the floor. **g, h** The previously solved X-ray structures of the BUNV head (pdb: 6H3V; light blue) and SBV stalk (pdb: 6H3S; light green) domains[6] were fitted into our model. The glycosylation site N624 is indicated (purple circle). **i, j** A model of the local lattice arrangement of BUNV GPs, emphasizing the two C3 symmetry axes; one forming a tripod, and one in the floor region where three neighbouring tripods connect. Scale bars: b = 100 nm, d = 5 nm; e, i = 2 nm.

from what is known for class II fusion proteins of alphaviruses[7,11]. Many bunyaviruses require low pH to induce a fusogenic state and, for some, a high H⁺ concentration has been reported to be sufficient to establish post-fusion conformations in vitro and trigger interactions with liposomes[14–16]. Working with the model peribunyavirus BUNV, and the model nairovirus, Hazara virus (HAZV), we recently showed the current description of bunyavirus entry is over-simplistic. We showed that the concentration of K⁺ ions ([K⁺]) in endosomes, which is controlled by cellular K⁺ channels and increases as endosomes mature[17], peaking in late endosomes, is an important cue for BUNV and HAZV entry. By blocking K⁺ channels pharmacologically, we showed that disruption of endosomal K⁺ accumulation impeded BUNV infection by preventing endosomal release, and instead, viruses were trafficked to lysosomes, where they were degraded[18–20]. For HAZV, we further showed that elevated K⁺ alone (at pH 7.3) triggered conformational changes in the GP spikes and induced interactions with co-purified vesicles[21]. Additionally, it has recently been shown that disruption of the cellular K⁺ gradient using KCl or the K⁺ ionophore valinomycin inhibits infection by the peribunyaviruses LACV, Keystone virus and Germiston virus; and of the phenuivirus RVFV[22,23], suggesting that K⁺ is broadly required during bunyavirus entry.

Here, we used cryo-electron tomography (cryo-ET) to unravel the structure of the envelope of BUNV and to identify changes in the virion architecture at endosomal pH and [K⁺]. STA was performed to generate an average of the BUNV GPs that allowed us to fit published atomic models of specific regions of Gc[6], and modelling both GPs (Gn and Gc) using AlphaFold[24] allowed us to generate a full pseudo-atomic model of the BUNV GP envelope. Furthermore, STA of pH 6.3/K⁺ treated virions revealed drastic changes in the GPs, including an uncoupling of Gn-Gc hetero-hexamers. Consistent with these structural data, biochemical assays confirmed that the structural changes elicited by K⁺ are localised to the floor region of the GP spikes. pH 6.3/K⁺ treated viruses also interacted more readily with target membranes, suggesting these conditions prompt exposure of the fusion loops, and when treatment occurs in the presence of a target membrane, it can trigger fusion.

## Results

### An improved STA of the BUNV GP permits modelling of the Gc and Gn hetero-hexameric arrangement

To determine the arrangement of Gn and Gc in the BUNV spike by cryo-EM, WT BUNV was concentrated and purified as previously described[5,25] (Supplementary Fig. 1). Purified virions (in a pH 7.3/no K⁺ buffer) were then vitrified on cryo-EM grids. Cryo-ET tilt series were collected and 3D tomograms were reconstructed, revealing roughly spherical virions (~100 nm) in which the lipid bilayer with tripod-like spikes was clearly identifiable surrounding a core of tightly packed ribonucleoproteins (Fig. 1b), as previously described[5]. STA was used to align and average ~20,000 spikes, which confirmed their tripodal arrangement, resolved at ~16 Å (by gold-standard (GS)-FSC Supplementary Fig. 2a and Supplementary Table 1). In this average, both leaflets of the viral membrane ('M') were clearly defined at the base, with a GP spike projecting ~16 nm from the surface (Fig. 1c, d). Our STA showed distinguishable head, stalk and floor regions (Fig. 1e, f), and a previously unidentified density protruding from the floor but shielded by the tripod head (Fig. 1e white arrowheads). Fitting of the previously determined X-ray structures of the head (BUNV) and stalk (SBV) domains into our model (Fig. 1g, h) revealed a tight fit corresponding to cross-correlation values of 0.70 and 0.82 for the head and stalk domains, respectively. The fit of the head trimer allowed us to confirm the handedness of the average (Supplementary Fig. 2b, c). By exclusion, the floor domain must contain the conserved class II fusion domain of Gc and Gn, however, specific arrangements could not be obtained from this average (Fig. 1g, h and Supplementary Movie 1).

To further our understanding of the architecture of this floor domain, STA was performed focused on the floor region (Fig. 1i, j) and resulted in an average at a slightly improved resolution of ~13 Å, as determined by GS-FSC (Supplementary Fig. 2a). This average is thought to represent a hetero-hexameric assembly of a trimer of Gn and a trimer of the Gc floor region, which includes the fusion domains. In the floor domain, details were identified that were not resolved in the previously published STA[5], highlighting the interconnected organisation of the fusion domains (Fig. 2a–d). Three stalk domains were also resolved, which identify the connections to three independent tripods. A post-fusion structure of the fusion domain for the related peribunyavirus LACV has been recently solved (pdb: 7A57)[26]. LACV Gc exhibits a class II fusion domain and shares ~49% amino acid identity and ~69% amino acid similarity with the fusion domain of BUNV Gc. We therefore modelled a LACV pre-fusion domain, based on the differences between the pre-fusion and post-fusion conformations observed for other bunyavirus class II fusion domains, which in broad terms involve the rigid-body rearrangement of domain III (Supplementary Fig. 3)[9,14,27,28]. In addition, there was unoccupied density remaining in the floor region that could accommodate a trimer of Gn in the centre, allowing it to contact the Gc fusion domains and the viral membrane (Fig. 2e, f).

There are currently no Gn structures solved for any peribunyavirus, however, an AlphaFold prediction of a LACV Gn monomer was previously generated[7]. For Gc only partial structures are available: the BUNV head domain, SBV, LACV and OROV stalk domains, and the LACV (post-fusion) fusion domain. In order to interpret the STA map of the BUNV pre-fusion spike, we generated an AlphaFold model of the entire Gn-Gc dimeric complex (Fig. 3a, b)[24,29], which reached an overall high confidence score with an average pLDDT of 85 (predicted local distance difference test, which is a per-residue estimate of its confidence; scores of >70 are expected to be modelled well[24]) (pLDDT scores and predicted aligned error plot in Supplementary Fig. 4a–c) with all conserved cysteines forming plausible disulphides. Fitting of the ectodomains of Gn and Gc from the AlphaFold model into our STA of the tripod (Fig. 3c, d, cross-correlation 0.72) required only minor adjustments of the angles between head/stalk and floor domains (Supplementary Fig. 4a, d; orange arrows), and it resulted in only minor clashes at protein contact sites within the floor.

At the time the prediction was made, AlphaFold had not yet been trained on the recently published post-fusion structure of the orthobunyavirus LACV Gc fusion ectodomains (pdb:7A57)[26], a circumstance that allowed further validation of the model. Indeed, the individual Gc fusion domains I, II, and III of the model are virtually identical to those of the LACV X-ray structure with RMSD values of 0.867-0.993 Å (pruned atom pairs; Supplementary Fig. 4e), and in the model are arranged in the canonical pre-fusion orientations relative to one another. Similarly, comparison of the AlphaFold model with the published BUNV head and SBV stalk domains revealed RMSD values of 0.419 Å and 1.087 Å respectively. The arrangement of the Gc fusion domain in the floor STA shows that a trimer fits well within this average (Fig. 3e, f; cross-correlation 0.73), and in a highly similar way to that predicted using the LACV structure with the fusion loops pointing upward underneath the tripod (Fig. 2e, f and Supplementary Movie 1).

The Gn AlphaFold model also displays the typical architecture of its orthologs from hantaviruses, phleboviruses, tospoviruses and alphaviruses, which are typically divided into four domains (domains A, B and C and the β-ribbon)[7]. BUNV Gn lacks the canonical domain B, but possesses domains A, C and β-ribbon observed in other bunyavirus Gn proteins[7], and similarly sits parallel to domain II of the Gc fusion domain, forming numerous stabilising interactions (Fig. 4a–c). The absence of domain B makes orthobunyavirus Gn significantly shorter than its orthologues, but this appears to be compensated by the head and stalk domains on Gc, which are not found in any other

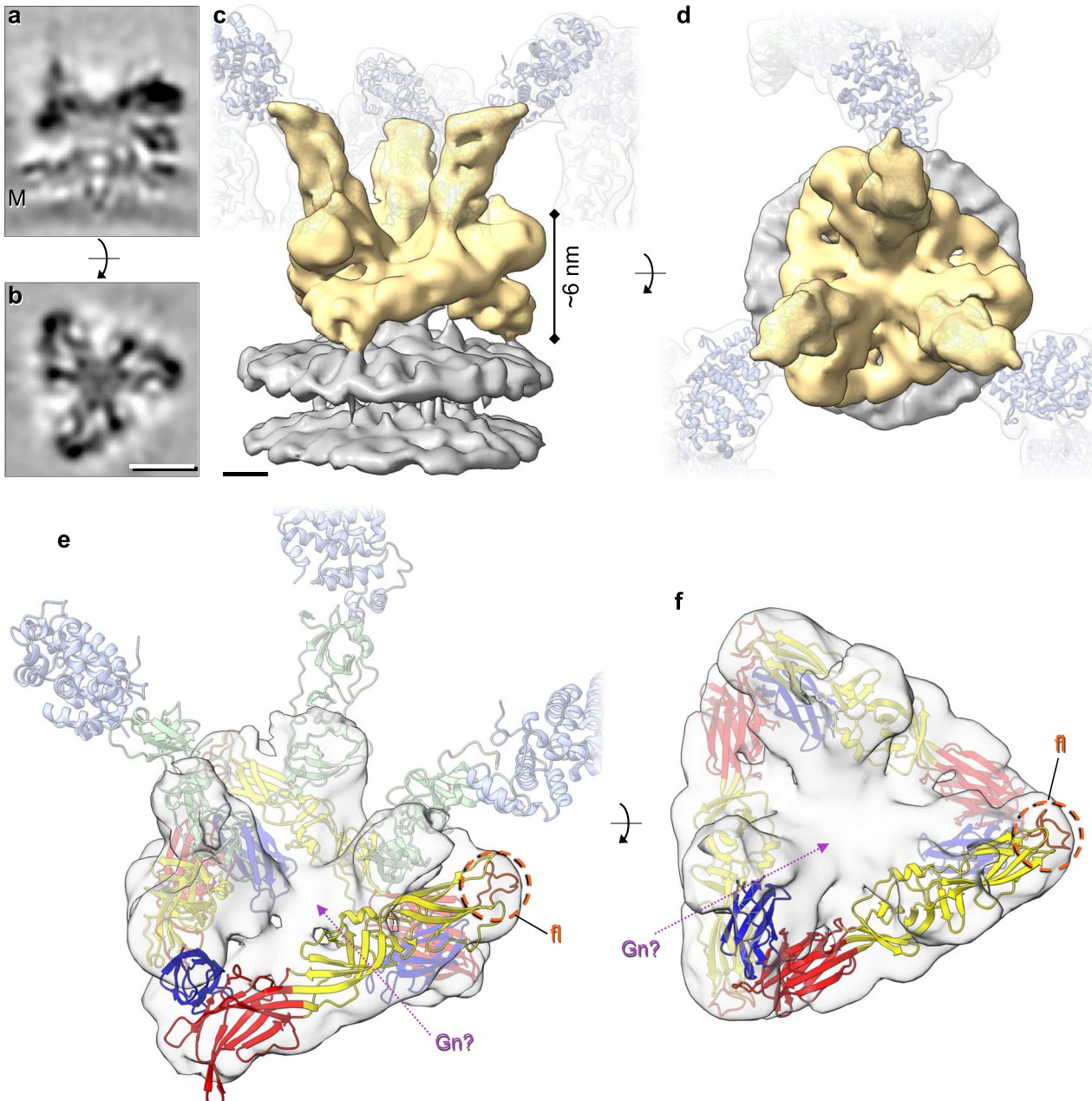

**Fig. 2 | STA of the floor region allows modelling of the BUNV fusion domain.** STA was performed as in Fig. 1, however aligning the floor region trimer (as opposed to the tripod). Over 16,000 subtomograms were aligned resulting in a ~13 Å resolution average (GS-FSC). **a**, **b** Sections through the electron density of the floor region average atop the viral membrane (M). **c**, **d** Isosurface rendering of the electron density (**c** side view, **d** top view) of the floor region (gold) atop the viral membrane (grey). Translucent tripods of the head-stalk (blue-green) regions have been added for orientation. **e**, **f** Modelling of the LACV fusion domain (residues 949–1344; domain I red, domain II yellow, domain III blue) in C3 symmetry within the BUNV floor region, revealing the location of the fusion loops (fl; orange) and remaining density that could be occupied by Gn (Gn?; pink arrow) (post-fusion structure pdb: 7A57). The location of the head (light blue) and stalk (light green) domains are indicated in (**e**) for orientation. Scale bars: **a**, **b** = 5 nm; **c**, **d** = 2 nm.

bunyaviruses (Fig. 4, compare a, b, c). Our unambiguous fit of three Gn-Gc heterodimer models into the triangular floor of the spike also demonstrates that three protomers of Gn oligomerize at the centre of the floor in a similar fashion as in the tetrameric hantavirus spike[9] (Fig. 4h).

Overall, the AlphaFold prediction confirms the orientation and positions of both Gn and Gc, and allowed us to model the whole orthobunyavirus GP envelope. This arrangement emphasises the role of Gn in stabilising the tripod and the fusion domains in the floor region and demonstrates how the lattice-like local arrangements are formed (Fig. 3g, h and Supplementary Movie 2).

## Acidic pH and K$^+$ alter the architecture of BUNV virions preventing virion aggregation

We are interested in the structural and functional changes of bunyavirus GPs in response to biochemical cues during entry. In our previous studies, we showed that the pH and K$^+$ dependence of virus entry can be recapitulated in vitro (termed acid-treatment), where exposing virions to physiologically relevant endosomal [H$^+$] and [K$^+$], at 20 mM or higher, led to accelerated onset of infection[19]. Here, we aimed to untangle the effects of low pH and K$^+$. Of note, when treating viruses in vitro at pH 6.3 prior to infection, the addition of K$^+$ resulted in a dramatic increase in the amount of viral N protein (a known marker for

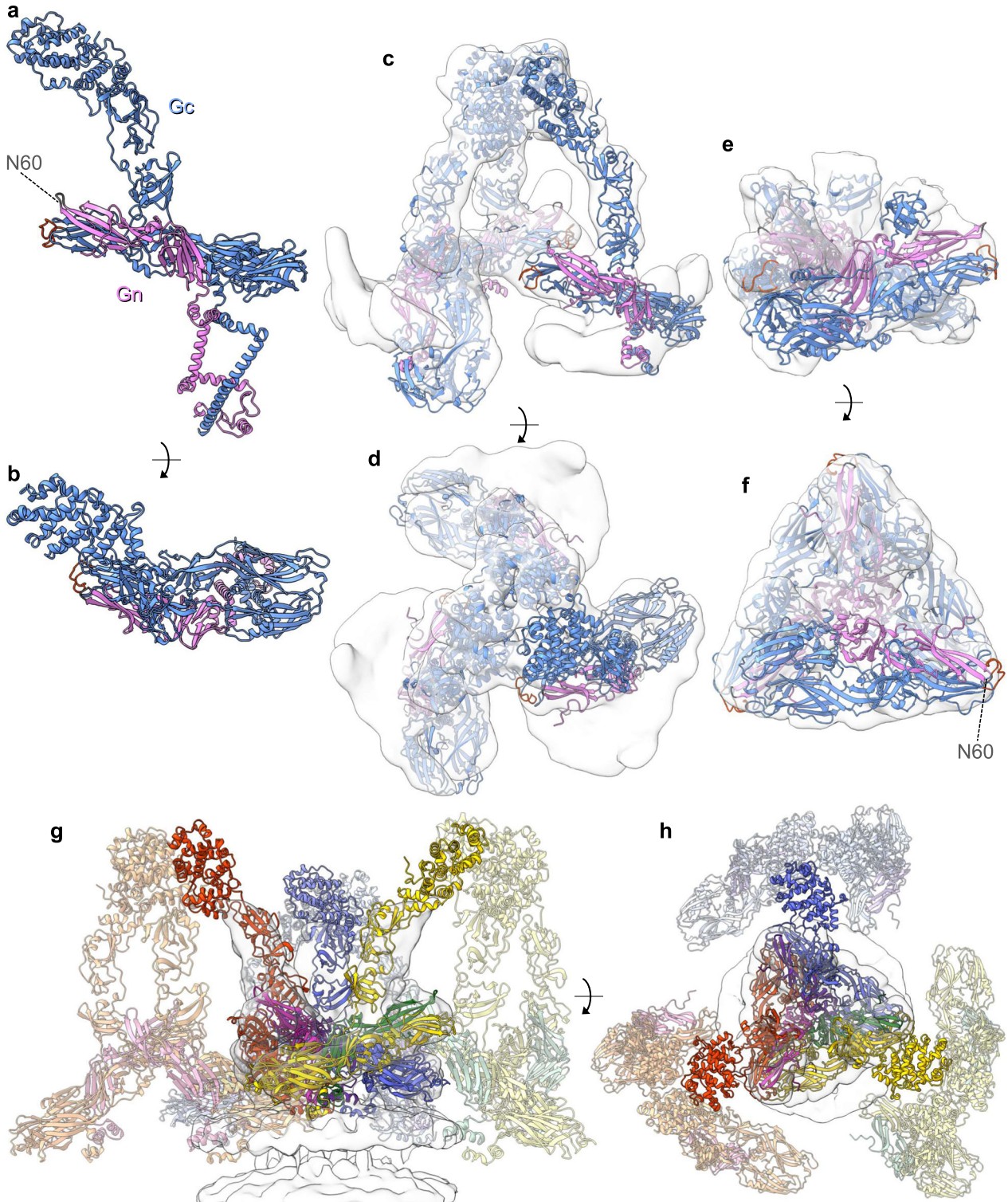

**Fig. 3 | AlphaFold dimeric structure of BUNV Gn and Gc fits reliably within our GP STA. a, b** AlphaFold generated an atomic model of the full-length Gn-Gc dimer. Gc in blue (fusion loop in orange), and Gn in pink with the N60 glycan on the Gn capping loop is highlighted in grey. **c, d** With only a minor rotation at an inter-domain region (Supplementary Fig. 4a, d), this model fitted well within the tripod STA (from Fig. 1; AlphaFold pdb found in Supplementary Data 1) in C3 symmetry (Gn-Gc ectodomains are shown; residues Gn 17-205, Gc 478-1382; no TMD). **e, f** The Gn ectodomain and the Gc fusion domain with the stalk subdomain II (residues 351–899) additionally fitted well within the floor region STA (from Fig. 2) in C3 symmetry. **g, h** A model of the BUNV GP envelope can be predicted, here showing three hetero-hexamers fitted within a floor STA. The Gn-Gc pairings are colour coded as follow; hexamer 1 purple-blue, hexamer 2 green-yellow, hexamer 3 pink-red. Supplementary Movie 2 also demonstrates this envelope model.

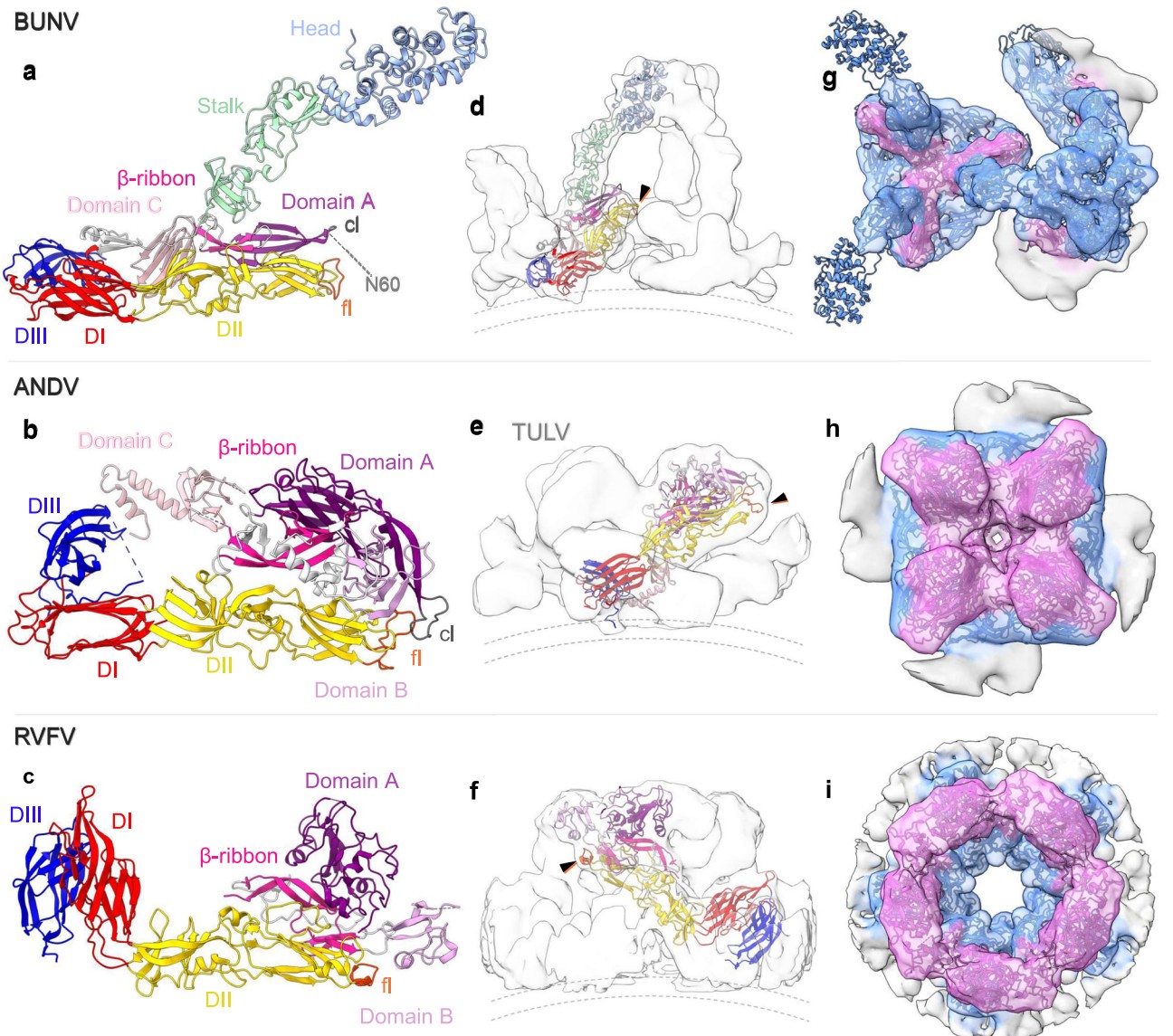

**Fig. 4 | Comparison of GP arrangements of BUNV with that of hantaviruses and phenuiviruses. a** The BUNV (orthobunyavirus) AlphaFold structure (as in Fig. 3) of Gc and Gn, as compared with the X-ray crystal structures of **b** the hantavirus ANDV (pdb: 6zjm) and **c** the phenuivirus RVFV (in the conformation as fitted within the EM average; pdb: 6f9f). The individual domain colours are the same for the three models, corresponding to the sub-domains of Gc (head (light blue), stalk (light green), domain I (DI, red), DII (yellow), DIII (dark blue) and the fusion loop (fl; orange)) and Gn (Domain A (dark-pink), Domain B (medium-pink), Domain C (light-pink), β-ribbon (magenta) and the capping loop (cl; grey)); as previously described[7,11]. White corresponds to the inter-domain regions. **d–f** Fit of a single Gn-Gc dimer within the corresponding EM density average; side view. The black arrows indicate the position of the fusion loop and dotted lines the viral membrane. **d** Shows the BUNV AlphaFold fit within the floor and tripod STA (as in Fig. 3). **e** ANDV Gn-Gc dimer within TULV STA reconstruction (EMD-11236)[9]. **f** RVFV Gn-Gc dimer within the EM single-particle reconstruction (EMD-4201)[8]. **g–i** Top-down view of the EM density averages with fitting of the Gn-Gc multimers: **g** BUNV trimer (within the floor and the tripod), **h** ANDV tetramer, **i** RVFV pentamer. The density for each has been coloured by distance from residues of Gn (pink) or Gc (blue), demonstrating the relative positions of each within the complex.

viral infection) observed at 18 hpi (during the exponential growth phase), and suggested a distinct role for K+ in expediting BUNV entry[19].

In order to characterise the structural effects of pH and K+ on BUNV GPs, we treated purified virions as above at both pH 7.3 and pH 6.3, and pH 6.3 with K+, prior to vitrification. 2D cryo-EM micrographs of pH 6.3/no K+ treated virions versus pH 7.3/no K+, showed that the morphology of the virions was altered and ~98% virions were forming clusters (compared to ~4% and ~13% of pH 7.3/no K+ and pH 6.3/K+ treated virions respectively, Fig. 5a, b; low magnification micrographs in Supplementary Fig. 5a, b), suggestive of direct interactions between virus particles. The electron-dense interior of individual virions can still be distinguished from one another, indicating virion-virion interactions are not fusion events. These interactions may have been driven

by exposure of the fusion loops of Gc on one virion contacting neighbouring virions, leading to clustering. Intriguingly, virions at pH 6.3/K+ formed no such clusters and appeared as individual virions (Fig. 5c), suggesting that K+ prevents the direct interaction of GPs with other viruses. Taken together, these data suggest that K+ has a direct effect on virion morphology and GP architecture and function, beyond that elicited by acidic pH.

**Cryo-ET reveals an altered GP arrangement in pH 6.3/K+ virions**

We next employed cryo-ET to characterise the morphology of virions treated in conditions mimicking early endosomes (pH 6.3/K+)[30]. Compared to pH 7.3/no K+ virions (Fig. 1b), pH 6.3/K+ virions were noticeably more pleomorphic, with clear disruption of the regular GP

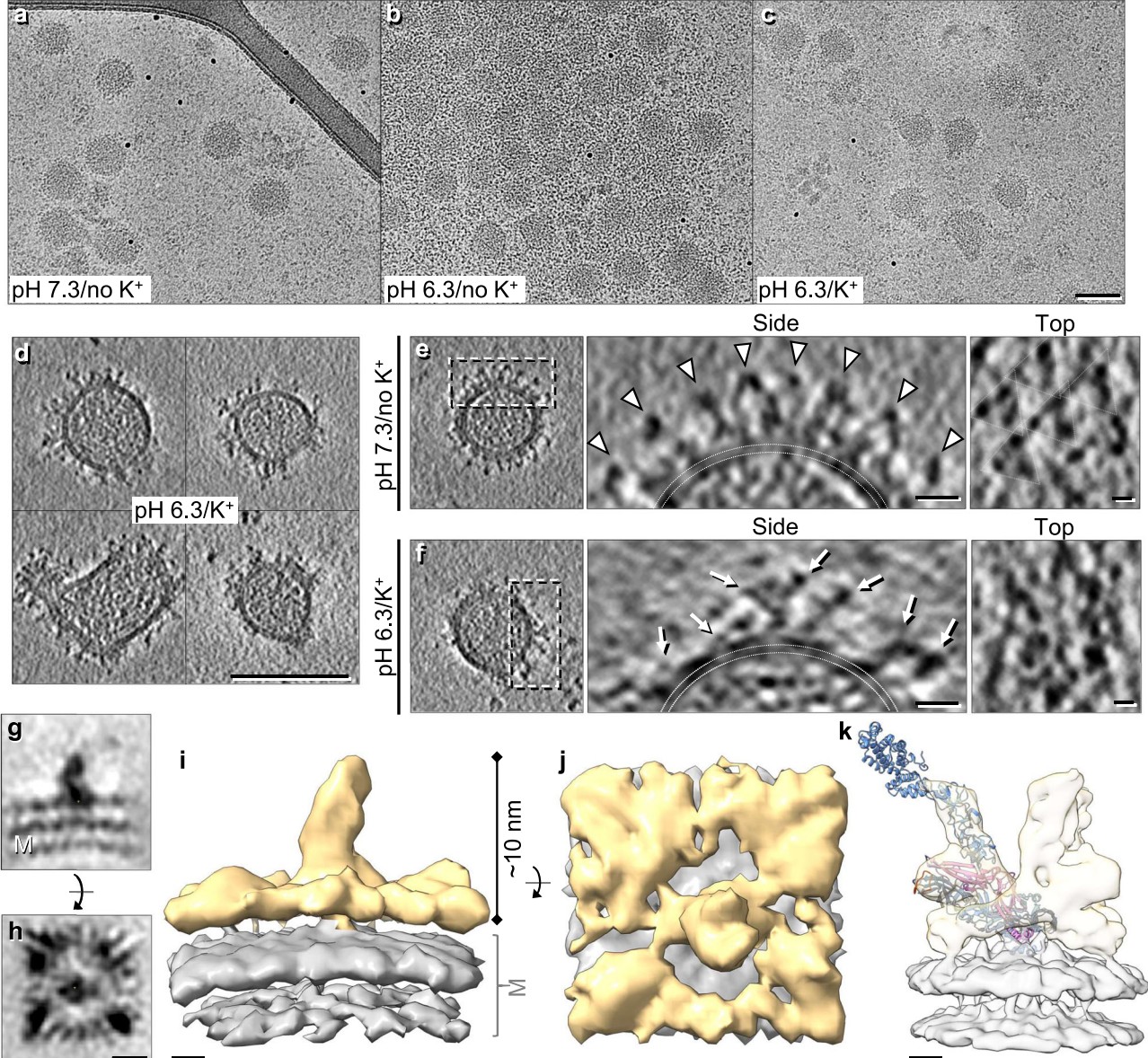

**Fig. 5 | pH 6.3/K⁺ disrupts the BUNV GP spike arrangements on the virion surface. a–c** Cryo-EM micrographs of BUNV virions treated under different conditions prior to plunge freezing: **a** pH 7.3/no K⁺, **b** pH 6.3/no K⁺ and **c** pH 6.3/K⁺ (*n* = 2, >10 micrographs per condition; low magnification images: Supplementary Fig. 5a–c) **d** Cryo-ET tilt series were collected of pH 6.3/K⁺ treated virions, then 3D tomographic reconstructions were calculated as with Fig. 1 (14 tomograms). Panels are slices through tomograms of pH 6.3/K⁺ treated virions which show clear disruption to the regular arrangement of GPs (representative of 89 virions, full tomogram example: Supplementary Fig. 5e). **e, f** An individual virion treated at **e** pH 7.3/no K⁺ (from Fig. 1) and **f** pH 6.3/K⁺ are shown alongside a slice through the side

(from the white dashed box) and top of each virion. White arrowheads in e emphasise the trimeric architecture and local lattice, and white arrows indicate spike 'ends' that no longer appear connected in tripods. **g–j** STA of ~10,000 subtomograms which were manually selected (owing to sample heterogeneity) from the surface of pH 6.3/K⁺ virions. **g, h** Slices through the centre of the spike electron density from the side (**g**) and top (**h**). The viral membrane is indicated (M). **i, j** Isosurface rendering of the density in g,h, with the spike (gold) on top of the viral membrane (grey). **k** Scaled comparison of the pH 7.3/no K⁺ floor region STA with the AlphaFold model fitted. Scale bars: **a–d** = 100 nm; **e**, **f** = 10 nm; **g, h** = 5 nm; **i–k** = 2 nm.

arrangement (Fig. 5d, f). Areas of GP-free membrane were also evident on ~48% virions. These data suggested that pH 6.3/K⁺ induced disruption to the locally ordered GP arrangements and allowed GPs to move more freely, revealing spike-free membrane regions.

In our tomograms, both the tripodal structure and local trimeric lattice were easily identifiable on pH 7.3/no K⁺ virions (Fig. 5e). However, pH 6.3/K⁺ virions showed a clear disruption of this arrangement and minimal evidence of tripods (Fig. 5f and Supplementary Fig. 5g), resembling the disruption previously observed in 2D cryo-EM at pH 5.1[5]. Of note, there were no obvious differences in the cores of the virions, suggesting that the effect(s) of pH 6.3/K⁺ were limited to viral GPs. Taken together, these data confirm that

treatment at pH 6.3/K⁺ leads to drastic changes in the GP structure of BUNV.

## pH 6.3/K⁺ results in GP hetero-hexamer uncoupling, dissociating the tripod and floor domain trimers

We subsequently used STA to determine the GP arrangement from pH 6.3/K⁺ virions. A dramatic change in GP architecture was observed, with the lack of clear tripods and the flexibility of the spikes evident early during processing (Fig. 5d, f, g–j). A short spike-like projection was aligned extending from the membrane (~10 nm), with a large, flat region of density above the lipid bilayer, which did not form the regular floor domain structure observed in the pH 7.3/no K⁺ condition.

The lack of density at the top of the spike was indicative of a highly dynamic unstructured region, suggesting multiple conformations as the density cannot be well refined by STA (Fig. 5f, g). However, two features were apparent from these averages: 1) the lack of organised tripods indicates uncoupling of this region, with the spike potentially corresponding to an extended Gc fusion domain (with or without Gn) or a stalk domain from the pH 7.3/no K$^+$ average (Supplementary Fig. 6b–d); 2) a floor region was present in this average, albeit it did not appear to form the ordered trimeric arrangement at the base of the spike comparable to the pH 7.3/no K$^+$ (Fig. 5k). Although some of the pH 6.3/K$^+$ densities might be visually similar to pH 7.3/no K$^+$ floor-like triangular conformations when viewed across a Z plane, the heterogeneity is still evident and any remaining symmetric floor domains are averaged out during processing indicating they are not a highly common occurrence.

In order to obtain further indirect information about the GP arrangement on pH 6.3/K$^+$ virions, we performed neutralisation tests with a monoclonal anti-Gc neutralising antibody (mAb-742) which recognises a conformational epitope on the head domain (near glycosylation site N624, which resides on the top of the head (Fig. 1a (asterisk) and h, N624))[31,32]. Western blot for BUNV-N of lysates from cells infected with viruses treated at pH 7.3/no K$^+$, pH 6.3/no K$^+$ or pH 6.3/K$^+$, and incubated with mAb-742, showed that this antibody neutralises infection by all treated virions (Supplementary Fig. 7a–d). This confirms that the head-stalk domains are intact and remain attached after pH 6.3/K$^+$ treatment, which was unclear from the STA. It is likely that the pH 6.3/K$^+$ viruses are still neutralised owing to steric hindrance, impeding the refolding of BUNV GPs or blocking the fusion loops from physically reaching the target membrane.

We additionally confirmed effects of K$^+$, beyond that elicited by H$^+$, in the floor region utilising a recombinant BUNV lacking the Gc head and ~70% of the stalk domains, which are replaced with eGFP (termed GFP-BUNV; Supplementary Fig. 7e)[33]. We previously described that for WT BUNV pre-treatment with pH 6.3/K$^+$ resulted in increased levels of BUNV-N at ~18 hpi compared to no treatment, suggesting that pH 6.3/K$^+$ expedited BUNV entry. Levels of pH 6.3/K$^+$ and pH 7.3/no K$^+$ BUNV-N expression, however, normalise by ~24 hpi, during the post-exponential plateau phase[19]. Here we confirmed that the GFP-BUNV mutant virus can be triggered by pH 6.3/K$^+$ and expedite infection similar to the pH 6.3/K$^+$ treated WT BUNV (Supplementary Fig. 7f, g), as previously described for WT[19]. Therefore K$^+$ must act on the floor region (i.e., Gn and/or the fusion domain of Gc), as this is the only common component between GPs from WT BUNV and GFP-BUNV. This aligns with our observations that pH 6.3/K$^+$ treated virions (Fig. 5f–j) lack an ordered arrangement in the GP floor domain. Together with the previous finding that the C-terminal portion of Gc is essential for virus infection[34], this indicates that the floor domain of the envelope is the main determinant for virus entry in vitro.

## pH 6.3/K$^+$ exposes the fusion loops and in the presence of a target membrane can induce fusion

As the GPs are key mediators of multiple stages of viral entry, we modified the in vitro pre-infection acid-treatment assays[19] to assess the effects of K$^+$ treatment on GP interactions with host membranes. Virions were in vitro treated at pH 6.3/K$^+$ ( + ), pH 6.3/no K$^+$ (-) or at pH 7.3/no K$^+$ as a control (pH 7), and the acidic buffers were diluted out prior to binding to cells at 4 °C (to block endocytosis). Any weakly bound virions were then removed by washing, and virus binding to host cells was assessed by measuring the amount of N at 24 hpi. For WT BUNV, PBS or trypsin washes efficiently remove surface-bound virions (Supplementary Fig. 8a). In the pH 7 control and pH 6.3/no K$^+$ treated viruses, little-to-no BUNV-N was detectable at 24 hpi when virions were washed after binding at 4 °C, indicating that virions treated at these conditions do not bind strongly to the cell surface (Supplementary Fig. 8b, c lanes 1–2, and 8d). In contrast, BUNV-N was detected at

similar levels to the unwashed viruses when virions were pre-treated at pH 6.3/K$^+$, indicating that these virions bind more strongly to host cell membranes and/or receptors, and are less readily removed by washing (compare Supplementary Fig. 8c, lane 3–4, 5, and 8d). As we showed that pH 6.3/no K$^+$ virions form clusters (Fig. 5b), we explored if clusters remain bound to the cell surface after washing. Bound virions (at an MOI of ~10) were fixed post-washing (0 hpi), then immuno-fluorescently labelled for BUNV-N (Supplementary Fig. 8e). This allowed us to confirm that large virion clusters are still bound, however, are likely unable to infect cells efficiently owing to this clustering.

We hypothesised that pH 6.3/no K$^+$ and pH 6.3/K$^+$ elicit structural changes that expose the previously shielded Gc fusion loops which facilitate contact with target membranes upon subsequent addition to cells, hence improving cell binding. Therefore, we next investigated whether these conditions are sufficient to induce fusion. For this we tested the ability of un-treated virions to fuse with the plasma membrane under acidic conditions using acid-bypass assays[35]. Virions were bound to cells at 4 °C, after which pH 6.3/no K$^+$, pH 6.3/K$^+$ or pH 5.3/no K$^+$ acidic buffers were added for a 2 min pulse to induce fusion at the plasma membrane and compared to an untreated control, which should not fuse. Post-bypass, pharmacological inhibitors were added to inactivate any remaining surface-bound virions (TCEP) and prevent endocytic entry (NH$_4$Cl)[19,20] (Fig. 6a). Finally, infection was assessed at 24 hpi. Under these conditions, any BUNV-N signal should represent successful fusion at the plasma membrane (Fig. 6b, lanes 1-5). Of note, BUNV-N signal was detected for the acid-bypass at both pH 6.3/no K$^+$ (lane 3) and pH 6.3/K$^+$ (lane 4) that was not present in the untreated control or pH 5.3/no K$^+$ conditions, indicating that these virions can bypass endocytosis and fuse at the cell membrane. The pH 6.3/K$^+$ BUNV-N fusion band (lane 4) was also the strongest of these conditions, suggesting that although pH 6.3/no K$^+$ can induce fusion pH 6.3/K$^+$ is more efficient at doing so.

This was also compared to infections in the same conditions as above, but without the pharmacological inhibitors, allowing, therefore, viral entry by both fusion at the plasma membrane and by endocytosis. As a result, all corresponding BUNV-N bands are stronger (Fig. 6b, c). The limited N-band in the untreated control (lane 6) is likely owing to the short duration of virus entry during the 2 min pulse, after which non-internalised viruses are removed; and the relative higher intensity of the bands in the adjacent lanes during the western blot exposure.

To confirm that pH 6.3/K$^+$ is inducing fusion when a target membrane is present, we incubated purified BUNV virions with artificial membranes, liposomes, at pH 7.3/no K$^+$ or pH 6.3/K$^+$ (37 °C, 2 hrs) to visualize fusion events by cryo-ET (similar to that previously described for influenza virus fusion[30]). After pH 7.3/no K$^+$ treatment, distinct virions could be seen in close contact with liposomes (Fig. 6d and Supplementary Fig. 9a, b and Supplementary Movie 3), which are identifiable owing to their electron-dense vRNP-containing core (as in Figs. 1b and 5a). After pH 6.3/K$^+$ treatment, fusion successfully occurred, as no distinct virions could be identified amongst the liposome clusters (Fig. 6e–h). Additionally, GP spikes could now be identified on liposomal membranes (black arrows) and vRNPs could be located within and surrounding liposomes (white arrows; free-vRNPs are likely due to liposomes breaking, either during the incubation or the freezing processes). This virus-liposome fusion is also exemplified by tomographic segmentation and 3D representation, where vRNPs can be identified within a spike-containing liposome suggesting that a virion(s) had fused with the liposome, releasing its contents within (Fig. 6h and Supplementary Movie 4). This suggests that pH 6.3/K$^+$ and a target membrane are sufficient to induce fusion. Of note, 2D cryo-EM projections of pH 6.3/no K$^+$ and pH 5.3/no K$^+$ conditions also showed no evidence of whole virions reminiscent of the pH 6.3/K$^+$ condition, suggesting

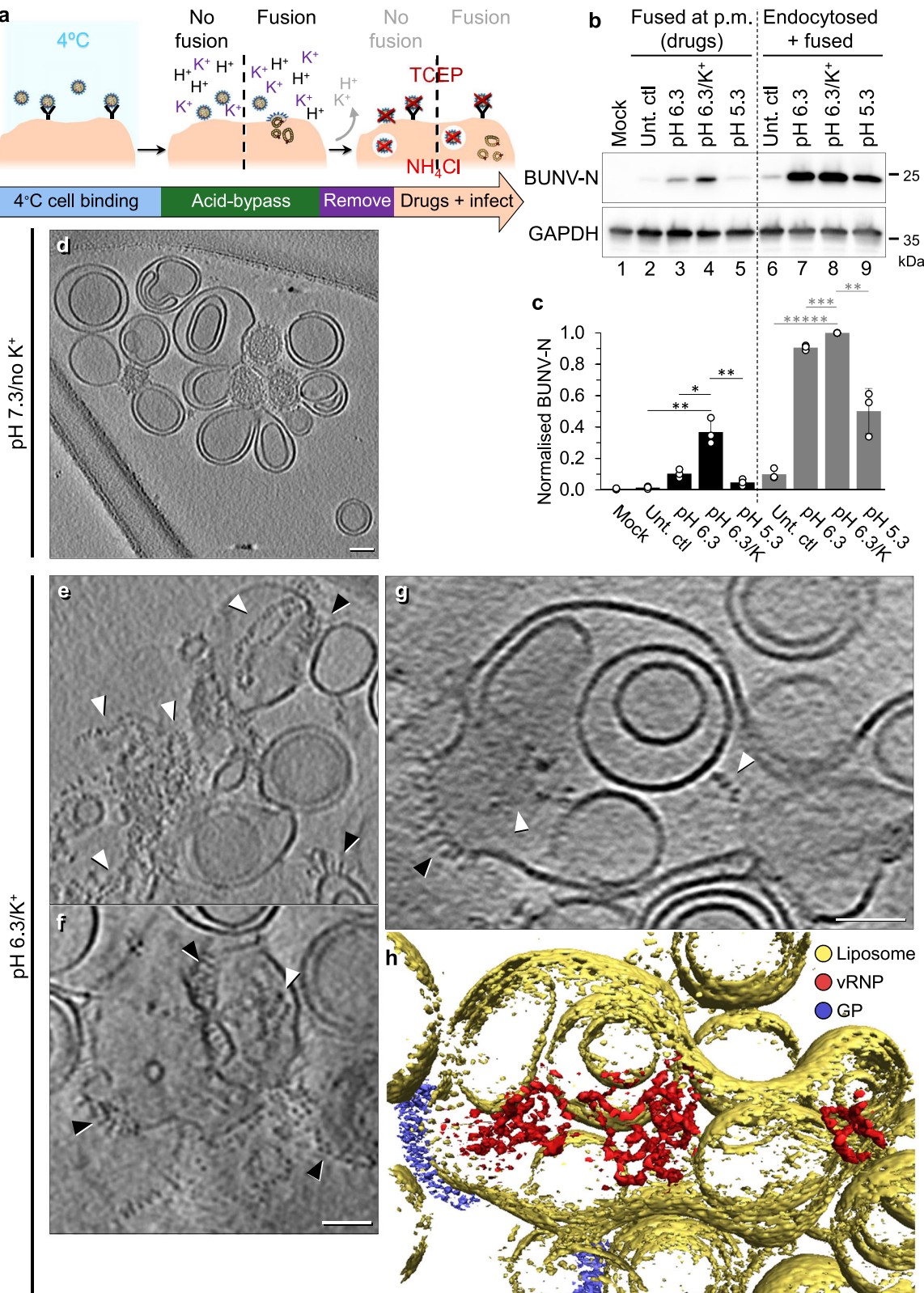

fusion had occurred (Supplementary Fig. 9a), however as pH 6.3/K$^+$ was ~3-fold stronger than without K$^+$ at inducing fusion (Fig. 6a–c) this was the condition taken forward to cryo-ET.

Together, these data indicate that K$^+$ and pH 6.3 act upon the BUNV GPs to promote conformational changes that increase interactions with membranes without fusion (Fig. 7b), however pH 6.3/K$^+$ in

the presence of a target membrane can induce fusion (Fig. 7a) and is the most efficient of the conditions tested.

## Discussion

We previously demonstrated that for BUNV low pH and K$^+$ treatment, and for HAZV neutral pH and K$^+$ treatment expedite infection[19,21].

**Fig. 6 | pH 6.3/K⁺ induces fusion in the presence of a target membrane.**
**a** Workflow of an acid-bypass assay. Virions are bound to A549 cells at 4 ˚C and fusion at the plasma membrane is attempted by acid-bypass at pH 6.3 (no K⁺), pH 6.3/K⁺, or pH 5.3 (no K⁺) for 2 mins at 37 ˚C. The addition of media for the 2 min 'bypass' was used for the untreated controls ('Unt. ctl'). Bypass buffer is replaced with media with or without drugs to prevent endocytic entry (TCEP inactivates virions outside the cell; and $NH_4Cl$ inhibits endosomal acidification). Cells are lysed at 24 hpi and assessed by western blot using antibodies against BUNV-N (a marker of infection) or GAPDH (loading control). **b** Western blot of experiment in (**a**), where BUNV-N presence in lanes 2–5 (black) indicates successful fusion at the plasma membrane (p.m.) as drugs to inhibit further endocytic entry are present. In lanes 6–9 (grey) BUNV-N presence represents fusion at the plasma membrane and endocytic entry, as infection is allowed to proceed post-bypass (no drugs or harsh washing steps performed) ($n = 3$). **c** Densitometry of $n = 3$ western blots in (**b**),

normalised to each loading control and then to the Unt. ctl (endocytosed+fused) (lane 6) (individual data points – white spheres). Error bars indicate mean ± standard deviation (SD), and significant difference determined using a one-way ANOVA comparing the significance of pH 6.3/K⁺ to each condition in the fused or endocytosed+fused (fused: Unt ctl $P = 0.002$, pH 6.3 $P = 0.006$, pH 5.3 $P = 0.003$. endocytosed+fused: Unt. ctl $P = 1 \times 10^{-6}$, pH 6.3 $P = 0.0004$, pH 5.3 $P = 0.004$).
**d**–**f** Purified BUNV virions were mixed with liposomes for 2 hrs at 37 ˚C (as in Fig. 5) at **d** pH 7.3/no K⁺, or **e**–**h** pH 6.3/K⁺, and then samples were vitrified on cryo-EM grids ($n = 3$). Cryo-ET tilt series were collected and 3D tomographic reconstructions were calculated as with Fig. 1. In **e**–**g**, white arrowheads indicate vRNPs, and black arrowheads viral GPs attached to liposomes (low magnification tomograms in Supplementary Fig. 9b, c). Scale bars = 50 nm. **h** Tomogram segmentation of the pH 6.3/K⁺ tomogram in (**g**) was performed in Amira. Liposomes = yellow, vRNPs = red and GPs = blue (also in Supplementary Fig. 9c and Movie 4).

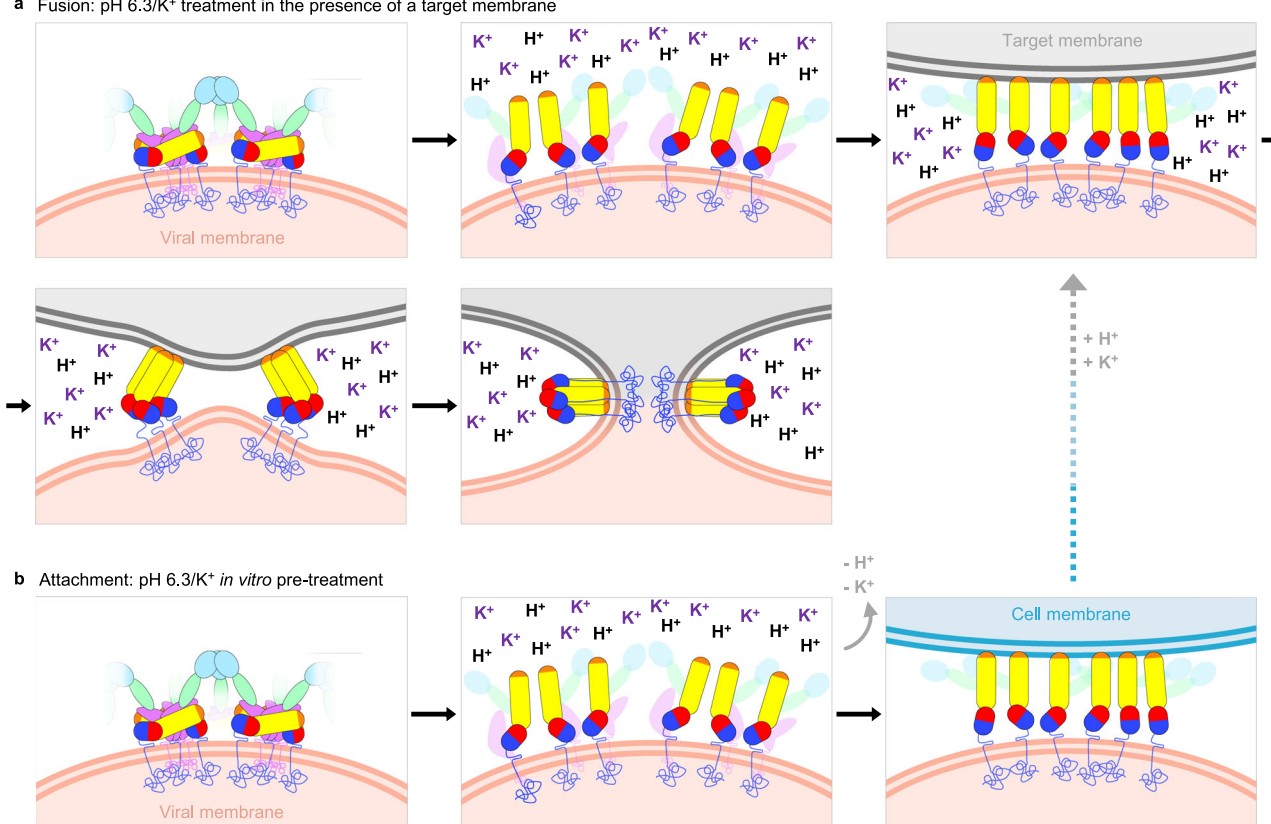

**a** Fusion: pH 6.3/K⁺ treatment in the presence of a target membrane

**b** Attachment: pH 6.3/K⁺ *in vitro* pre-treatment

**Fig. 7 | Model hypothesising the role of pH 6.3/K⁺ in BUNV fusion events. a** pH 6.3/K⁺ in the presence of a target membrane (liposome, endosomal or plasma membrane) is sufficient to induce fusion. This is based on results from the acid-bypass assay, where the target membrane is the plasma membrane (Fig. 6a–c) and on the virus-liposome fusion observed by cryo-ET (Fig. 6e–h). The canonical model of fusion and the structural changes in class II fusion proteins are multi-step[13]. Instigated by exposure to a fusion trigger (e.g., pH 6.3/K⁺ for BUNV), the pre-fusion conformation must first dissociate in order to release the previously shielded fusion loops, generating an extended intermediate structure which can now facilitate insertion into a target membrane. The structure of the head and stalk domains is unknown at this stage, but the fact that the anti-Gc neutralising antibody mAb-742 is effective after pH 6.3/K⁺ suggests the structures of the individual domains remain unchanged and they remain attached to the virion. Trimerization of the

fusion domains is then required for successful foldback of the fusion proteins to bring the two opposing membranes into close contact and overcome the kinetic barrier to fusion[53]. This then allows the two membranes to hemifuse and then fuse, involving further structural changes in the GP that lead to a post-fusion conformation, whereby the fusion loop and transmembrane domains are now inserted into the same membrane. **b** In the absence of a target membrane (e.g., during in vitro pH 6.3/K⁺ pre-treatment) the hetero-hexameric pre-fusion conformation disassembles, as seen in the cryo-ET and STA (Fig. 5f–j). This likely represents an irreversible extended intermediate conformation that is then able to insert into host membranes and bind cells more strongly (Supplementary Fig. 8c, d). Further re-introduction of H⁺ and K⁺ in the presence of a target membrane is required for fusion (as previously described[19]). All three conditions lead to productive fusion: target membrane, H⁺ and K⁺.

However, mechanistic insight was limited owing to the low resolution of the HAZV STA and lack of complete GP structures available for BUNV and HAZV. Here, we utilised BUNV to investigate the mechanistic and structural consequences of endosomal pH and K⁺ during BUNV entry using cryo-ET and STA.

First, we improved the average of the BUNV spike, allowing us to fit structures of the BUNV head and SBV stalk, confirming previous predictions that the fusion domain resides in the floor region (Figs. 1, 2)[5,6]. Our improved resolution of the floor region, coupled with an Alpha-Fold model of the complete BUNV Gn-Gc dimer, allowed us to model

the whole viral ectodomain and demonstrates the hetero-hexameric 3x(Gn/Gc) assembly (Fig. 3g, h). Additionally, the fitting of Gc in our subtomogram average shows by exclusion that Gn resides at the centre of the floor domain, connecting the stalks from three tripods, likely stabilising this region. This assembly resembles the organization of the Alphavirus envelope, in which three protomers of the class II fusion protein E1 laterally surround a trimeric core of its chaperone E2[36]. A similar assembly, with different stoichiometry, is also found in the hetero-octameric 4x(Gn-Gc) complex of the orthohantavirus envelope (Fig. 4e, h)[9]. In contrast to other bunyaviruses however, our model of the BUNV envelope illustrates that the orthobunyavirus Gn cannot completely cover the Gc fusion domain (Fig. 4g). The capping loop found at the tip of BUNV Gn is also shorter than that of hantaviruses, although it still partially shields the fusion loops (Fig. 4a, b). Additionally, a strictly conserved N-linked glycan on the capping loop (residue N60) is likely maintained to allow direct interaction with the fusion loops on Gc, as described for hantaviruses[9] (Figs. 3a, f and 4a). This BUNV Gn glycan was previously shown to be essential for correct GP folding and successful virus rescue, suggesting an important function in the regulation of the pre/post-fusion switch[32]. The unique N-terminal half of the larger orthobunyavirus Gc appears to compensate for the small size of its Gn, being well positioned for receptor scavenging and indirectly shielding the fusion machinery (Figs. 3, 4 and Supplementary Movie 1). These head and stalk domains of BUNV Gc are however dispensable for infection in vitro and are not involved in cell fusion or Golgi trafficking, indicating no role in other stages of infection[33,34]. Furthermore, our results show that the head and stalk domains are not required for virus triggering by pH 6.3/K⁺, indicating an effect of this condition on the floor region (Supplementary Fig. 7e–g).

With the shielding of the fusion loops, disassembly is required during fusion, similar to the glycoprotein rearrangement required for exposure of the fusion loops of RVFV, HTNV and ANDV[8,9,37,38]. As is established for class II fusion proteins[13], this pre-fusion conformation disassembly is initiated by a biochemical trigger during entry which would elicit structural changes in the GPs to expose the fusion loops, permitting their interaction with endosomal membranes. This is what we observed upon pH 6.3/K⁺ treatment, where pH 6.3/K⁺ triggers the disassembly of the organised hetero-hexameric arrangement, generating an intermediate state that likely involves exposure of the fusion loops to target membranes. This is consistent with the K⁺ effects previously shown for HAZV, where K⁺ treatment triggered large structural changes in the GPs at neutral pH, causing elongation of the GP spikes and their association with co-purified membranes[21]. At a more acidic pH 5.0, loss of the ordered tetrameric arrangement is also observed for TULV (hantavirus) GPs[38], and the elongation and membrane interaction of GP spikes of UUKV (phenuivirus)[39]. However, the requirement of K⁺ for TULV and UUKV entry has not been explored.

For BUNV, if virions are pH 6.3/K⁺ treated when a target membrane is present, fusion can occur, as is shown with the acid-bypass (plasma membrane; Fig. 6b, lane 4) and liposome-virus fusion (Fig. 6e–h) assays. These conditions are therefore sufficient to induce the subsequent structural rearrangements necessary for fusion (Fig. 7a), suggesting that no other host factors are required. In the absence of a target membrane (in vitro pre-treatment), disassembly of the ordered GP arrangement (Fig. 5f–j) establishes a conformation with enhanced binding to the cell post-treatment (Supplementary Fig. 8b, d), however does not lead to an irreversible post-fusion conformation, as these viruses are still fusion-capable and infectious post-treatment. In addition, our previous work determined that endosomal H⁺ and K⁺ are still required post-in vitro-treatment at pH 6.3/K⁺ but to a lesser extent[19], suggesting low pH and K⁺ are required for further structural changes in the fusion process beyond the initial disassembly (Fig. 7b, dotted line).

Interestingly, previous studies on the BUNV GP suggested the occurrence of uncoupling at pH 5.1 in the absence of K⁺[5], later shown to render BUNV non-infectious[19], therefore likely representing a post-fusion conformation. Fusion events have also been observed at pH 5.3 (no K⁺) using cell-cell fusion assays with BUNV GPs expressed on the surface of BHK cells[34,40]. Although in our 2 min acid-bypass assay pH 5.3/no K⁺ did not elicit fusion (Fig. 6a–c), likely due to the short incubation time, the 2 hr liposome fusion assay did suggest that this condition can actually induce fusion (Supplementary Fig. 9a).

We also demonstrated that although pH 6.3/K⁺ was the pre-eminent condition at inducing fusion, pH 6.3/no K⁺ can also induce fusion in acid-bypass and liposome assays, suggesting the H⁺ alone can induce fusion (Fig. 6a–c and Supplementary Fig. 9a). Compared to pH 6.3/K⁺, similar structural changes in BUNV may also be elicited at low pH alone (pH 6.3/no K⁺), which allows interactions with other virions (Fig. 5b). However, the lack of virus-virus interactions in the presence of K⁺ at pH 6.3 suggests that K⁺ limits the effects of pH and prevents full exposure of the fusion domains. Interestingly, a post-fusion X-ray structure of HTNV Gc at pH 6.5 in the presence of KCl coordinates a K⁺ ion within domain II, at tyrosine residue Y105, which then forms different interactions and a more unstable arrangement[41]. A similar scenario may be true for the pH 6.3/K⁺ BUNV Gc, in which a K⁺ ion coordination may affect exposure or stability of the fusion loops, thereby preventing virion aggregation as the fusion loops cannot interact with one another. The fact that we cannot resolve a stable conformation of the pH 6.3/K⁺ GP spike is also suggestive of an unstable, flexible conformation (Fig. 5f–j). Of note, for the orthobunyavirus Germiston virus, fusion can be induced at pH 6.0 and the endocytic switch from Na⁺ to K⁺ was also shown to be important for entry[23]. Although the effects of pH 6.0 with K⁺ on Germiston virus fusion were not investigated, this represents an interesting comparison and suggests a shared mechanism.

In summary, we have determined the molecular arrangement of the Gc fusion domain and Gn within the floor domain of the BUNV spike, and additionally resolved drastic structural changes in the GP architecture in response to pH 6.3/K⁺, which in the presence of a target membrane can induce fusion. The ability of pharmacological K⁺ channel inhibitors to disrupt infection by a number of bunyaviruses from different families[18,21,22,42], alongside the structural changes elicited by K⁺ on both BUNV and HAZV, which is potentially shared by other bunyaviruses, suggests a shared mechanism that could be exploited therapeutically.

## Methods
### Cells and viruses
A549 (human alveolar carcinoma epithelial; 86012804), SW13 (human adrenal cortex carcinoma; 87031801) and BHK-21 (baby hamster kidney; 85011433) cells were obtained from the European Collection of Cell Cultures (ECACC). Cell lines were authenticated by ECACC and were routinely tested for mycoplasma. Cells were cultured in Dulbecco's modified Eagle's medium (DMEM, Sigma) supplemented with 10% foetal bovine serum (FBS), 100 U/ml penicillin, and 100 μg/ml streptomycin, and maintained in a humidified incubator at 37 °C, with 5% $CO_2$. Wild-type BUNV stocks (provided by Professor Richard Elliott, Glasgow Centre for Virus Research (CVR)) were generated from infected cell supernatants and titre estimated by plaque assay[18,19]. For infection assays, A549 cells were infected at an MOI = 0.1 unless otherwise specified. GFP-BUNV stocks were provided by Dr Xiaohong Shi (Glasgow CVR) and encoded an eGFP tag replacing residues 478-827 of the BUNV M segment (the Gc head and stalk regions)[33]. Wild-type human respiratory syncytial virus A2 strain (HRSV) was obtained from the National Collection of Pathogenic Viruses (NCPV; Public Health England)[43].

## Virus purification

BHK-21 cells were seeded into T175 flasks and infected with BUNV for 3 hr in FBS-free DMEM (MOI of 0.1). Media was replaced with 2% FBS DMEM and incubated until 44 hpi at 32 °C. Virus-containing supernatants were clarified by centrifugation and filtration (0.22 μm filter), and then virions were pelleted by ultracentrifugation (150,000 x *g*, 3 hr, 4 °C) through a 30% sucrose cushion[25]. Virions were resuspended in 0.1x phosphate-buffered saline (PBS, 0.1x to dilute the salts) by gentle rocking at 4 °C overnight. BUNV titres were determined by plaque assay, and purity was determined by silver staining and negative stain EM.

## Plaque assays

Plaque assays were performed in SW13 cells as previously described[19]. Viral stocks were serially diluted from $10^{-1}$ to $10^{-6}$, and SW13 cells were infected for 1 hr at 37 °C. Infected cells were overlayed with 1.6% carboxy-methyl cellulose diluted equally in complete DMEM. After 6 days the overlay was removed, cells were fixed in 4% paraformaldehyde for 15 mins at 4 °C, stained with 1% crystal violet and the number of plaques counted to estimate viral titre.

## Silver staining

During the BUNV purification 10 μl samples were collected of the pre-ultracentrifugation viral supernatant (snt), the post-ultracentrifugation snt, snt-sucrose interface, and sucrose. Samples were resolved by SDS-PAGE alongside 0.5 μl of purified BUNV and 1:100 diluted protein ladder, then fixed and stained following the silver staining kit (GE Healthcare) manufacturer's instructions.

## Western blot

At the indicated time points cells were lysed (25 mM glycerol phosphate, 20 mM tris, 150 mM NaCl, 1 mM EDTA, 1% triton X-100, 10% glycerol, 50 mM NaF, 5 mM $Na_4O_7P_2$, pH 7.4), and western blotting was performed[19,43]. Proteins were resolved by SDS-PAGE and transferred onto polyvinylidene difluoride (PVDF; Millipore) membranes. After blocking in 10% (w/v) milk in TBS-T, the appropriate primary antibodies were added at 4 °C overnight: sheep α-BUNV-N (1:5000) produced by J N Barr, University of Leeds (antibody-containing serum was obtained from a sheep inoculated with BUNV-N protein previously expressed in E. coli and purified by affinity and size-exclusion chromatography detailed in Ariza *et al.* (2013)[44]; and used extensively in previous publications[18]); goat α-HRSV (1:1000; Abcam ab20745); mouse α-GAPDH (1:2000, Santa Cruz sc47724); or mouse α-GFP (1:1000; Santa Cruz sc-9996). This was followed by the corresponding HRP-conjugated secondary antibodies (Merck α-sheep A3415; α-goat A8919; α-mouse A4416) for 1 hr. Labelling was detected using enhanced chemiluminescence (Advansta) and a G:Box processor. Densitometry was performed using ImageJ, showing standard deviation error bars and statistical significance determined using a one-way ANOVA where P < 0.05. Full uncropped blots and densitometry data can be found in the Source Data file.

## Immunofluorescence staining

After the indicated infection times, cells were washed with PBS and then fixed for 15 mins at 4 °C with 4% paraformaldehyde. Fixed cells were permeabilised with 50% methanol/acetone, blocked in 1% bovine serum albumin (in PBS) for 30 mins and labelled using α-BUNV-N primary antibodies (1:5000) for 1 hr, followed by α-sheep Alexa-fluor-594 or -488 fluorescent secondary antibodies added 1:500 for 1 hr (Thermo Scientific A11016 and A11015 respectively). As specified, fluorescence was imaged using the IncuCyte ZOOM imaging system, or coverslips were mounted onto glass slides (ProLong Gold Anti-Fade reagent; Invitrogen) and imaged using a Zeiss Confocal LSM880 upright microscope (40x magnification).

## In vitro acid pre-treatment assays

BUNV acid pre-treatment assays were previously optimised, including assessment of a range of time points, ionic buffers and physiological KCl concentrations[19]. Therefore, these in vitro treatment assays are from established protocols and were performed as previously described. Any adaptations are highlighted. BUNV was incubated pre-infection at 37 °C for 2 hrs in a range of buffers: pH 7.3 (20 mM tris), pH 6.3 (30 mM bis-tris) or pH 5.3 (50 mM sodium citrate) supplemented with 10 mM NaCl (to maintain osmolarity), with or without 140 mM KCl ('K+' vs 'no K+'). Buffer was diluted out with DMEM prior to infection of A549 cells at a MOI of 0.1 for the indicated times (hpi), after which cells were lysed and BUNV infection assessed by western blot. Cell lysis at 18 hpi represents the exponential phase of BUNV infection (where pH 6.3/K+ expedited entry at BUNV-N expression begins earlier than WT and there is therefore more N) and 24 hpi during the plateau post-exponential phase (pH 6.3/K+ treated virions and WT BUNV-N expression is equal)[19]. For the GFP-BUNV, virus was treated with buffers as with WT BUNV, and infection was assessed at 18 hpi.

For the cell binding assays, after the pre-infection in vitro treatment (pH 6.3/no K+, pH 6.3/K+, or a pH 7.3 control), buffers were diluted out with cold DMEM and virions added to cells at 4 °C for 1.5 hrs; to allow binding but prevent internalisation[19]. Cells were washed 3x 30 secs with 0.1x PBS or left unwashed, and infection was allowed to proceed until 24 hpi (see Supplementary Fig. 8b). Infection was assessed by western blot of cells lysed at 24 hpi. In the immuno-fluorescence binding test, purified virions were in vitro treated (pH 7.3/no K+ or pH 6.3/K+), bound to cells at 4 °C for 1.5 hrs, and then washed as above. Cells were fixed (0 hpi) and immunofluorescently stained for BUNV-N. Purified BUNV of a high titre (~$1 \times 10^9$ PFU/ml) was used in order to bind viral particles an MOI of ~10 to the cells, to improve virion detection by this method. Images were taken by confocal microscopy. In the initial wash test assay (Supplementary Fig. 8a), A549 cells were infected at 4 °C with BUNV (MOI = 0.1) and left unwashed or washed 3x 30 seconds with 0.1x PBS or 0.1% trypsin. Infection was assessed at 24 hpi by western blot.

For the neutralisation assays, BUNV was treated at pH 7.3/no K+ (control), pH 6.3/no K+ or pH 6.3/K+ as above, then buffer was diluted and virions were neutralised with mAb-742 anti-BUNV-Gc monoclonal antibody (1:10,000; produced in the Professor Richard Elliott laboratory, CVR, University of Glasgow and provided by Dr Xiaohong Shi[31]) or a dH2O control for 1 hr with gentle rocking. A549 cells were then infected at an MOI 0.1 for 18 hrs, at which point cells were fixed for immunofluorescence or lysed for western blot. Fixed cells were immunofluorescently stained for BUNV-N and images taken using an IncuCyte ZOOM as previously utilised[19,43]. HRSV control experiments were performed as with WT BUNV, however infections were performed at an MOI = 1 for 24 hrs, and analysed by western blot.

## Acid-bypass assay

To induce fusion at the plasma membrane, and thus bypass endosomal entry (see Fig. 6a), a protocol was adapted from the procedure previously outlined by Stauffer *et al.* (2014)[35]. BUNV (MOI 0.1) was bound to A549 cells by addition at 4 °C for 1 hr, and unbound virus was then removed with the media. Virions bound to the cell membrane were subjected to acidic buffers ('acid-bypass') by adding warm fusion buffers to cells for a 2 min pulse at 37 °C: pH 6.3/no K+, pH 6.3/K+, pH 5.3/no K+, or a DMEM-added infection control (Inf. ctl) (instead of the 2 hrs used for acid pre-treatments). The fusion buffers used were the same as those used for the acid-treatments in the above experiments, but additionally containing 0.1x PBS to maintain cell osmolarity. Conditions were performed in duplicate and to one of each buffer condition, fusion buffers were gently removed and replaced with DMEM (no harsh washing steps to avoid removing weakly bound virions). Thus, endocytic entry, as well as plasma membrane fusion, will proceed (labelled: endocytosed+fused). To the duplicate well for each

fusion buffer condition, cells were subsequently washed with cold DMEM and treated with or without drugs to prevent endocytic entry: the reducing agent tris(2-carboxyethyl)phosphine (TCEP; 10 mM, Sigma) was added to cells for the + drugs wells for 5 mins at 37 °C to inactivate surface-bound virions[20], which was then removed and replaced with an acid-bypass stop buffer (DMEM containing 50 mM HEPES (pH 7.3), and 20 mM ammonium chloride (NH$_4$Cl); Sigma) to prevent endosomal acidification and hence entry by endocytosis[19]. As such, only virions that bypass the endocytic network and fuse at the plasma membrane would result in a productive infection[19,20] (labelled: fused at p.m. (drugs)). Cells were incubated at 37 °C for 24 hrs, then lysed and infection assessed by western blot.

## Liposome preparation
Liposomes consisting of phosphatidic acid, phosphatidylcholine, and rhodamine-labelled phosphatidylethanolamine (ratio 10:10:1) with 10% cholesterol were purified as previously described[45]. Briefly, lipids in chloroform were dried in a film with argon and rehydrated in 10 mM NaCl, 10 mM Hepes pH 7.5. The rehydrated lipids were initially extruded through a mini-extruder (Avanti Polar Lipids) using a 400 nm pore-size membrane (Whatman) and further extruded using a 100 nm pore-size membrane (Whatman). Liposomes were pelleted by ultracentrifugation at 140,100 $xg$ (20 °C) for 20 minutes using a SW55 rotor. A comparison of the rhodamine fluorescence in the liposome preparation and samples of rehydrated lipids with known lipid concentrations was used to estimate the lipid concentration of the liposomes.

## Electron microscopy
Negative stain EM was used to assess the purification of BUNV prior to cryo-EM[21]. Briefly, purified BUNV was loaded onto glow-discharged carbon-coated grids and allowed to stand for 30 secs. Grids were washed three times with dH$_2$O and stained with 1% aqueous uranyl acetate for 10 secs. Images were collected on a FEI Tecnai T12 electron microscope at 120 kV, using a Gatan Ultrascan 4000 charge-coupled device camera and operated between -1 μm and -5 μm nominal defocus.

For cryo-EM and cryo-ET, 2 μl purified BUNV virions ( ~ 6.8 x 10$^6$ particles) were diluted in 2 μl (1:1) of a range of acidic buffers (components outlined in 'acid pre-treatment' section above, however, adjusted to equate the same final concentrations: pH 7.3/no K$^+$, pH 6.3/no K$^+$, pH 6.3/K$^+$; final concentration 127.5 mM KCl) for 2 hr at 37 °C matching the acid-treatment experiments used previously[19]. Virions were then mixed with 2 μl of Protein A conjugated with 10-nm colloidal gold (Aurion) as a fiducial marker for tomogram alignment, and 3 μl of the mixture was immediately loaded onto glow-discharged lacey-carbon EM grids, with an ultra-thin 2 nm carbon support film (Agar scientific). Grids were blotted for 3 secs and vitrified using a Leica EM GP automatic plunge freezer. Cryo-EM micrographs were collected for each condition on an FEI Titan Krios using a Falcon E3C direct electron detector, operated at 300 KeV and at −0.5 to −3.5 μm defocus.

For the liposome fusion assay, 2 μl purified BUNV ( ~ 2.25 × 10$^6$ particles) were combined with 1 μl liposomes and 3 μl buffer (buffer pH 7.3/no K$^+$, pH 6.3/no K$^+$, pH 6.3 or pH 5.3/no K$^+$ as for cryo-EM of virions) for 2 hr at 37 °C, conditions known to elicit changes in the BUNV GPs. Samples were then cooled at 4 °C until 3 μl colloidal gold was added and immediately vitrified on cryo-EM grids, as above. Cryo-EM grid screening micrographs were collected as above.

## Cryo-ET and image processing
Grids for cryo-ET were prepared as described above and tilt series were collected using a FEI Titan Krios using an energy-filtered (20 eV slit) Gatan K2 XP summit direct electron detector in counting mode (300 KeV), and a Volta phase plate. Tomography 4 (FEI) and EPU v2.8.1 software were used to collect single-axis tilt series from −60° to +60°, at 2° increments. A defocus of -1 μm was used, at x53,000

magnification, corresponding to a pixel size of 2.72 Å, and an electron dose of -1.8 e$^−$ per image (total dose per tilt series -108 e$^−$/Å$^2$). Tilt-series projections were pre-processed by motion correction using Relion 3.0, and contrast transfer function (CTF) calculated using Relion 3.0 (gctf) and then corrected using CTF phase-flip[46,47]. The IMOD package eTOMO was used to align the projections using the gold fiducials and calculate the 3D reconstructions by weighted back projection[48], with a final pixel size of 5.44 Å after binning by a factor of two. For STA, unbinned tomograms (2.72 Å pixel size) were also generated using only the −30˚ to +30˚ tilt angles (to improve the signal-to-noise of the resulting tomograms). Representative images were Gaussian Blur 3D filtered using ImageJ.

Cryo-ET tilt series were collected for the liposome-virus fusion assays for the pH 7.3/no K$^+$ and pH 6.3/K$^+$ conditions, using similar parameters to those above (−60˚ to +60˚, at 2˚ increments) on a FEI Titan Krios with a Falcon 4i detector in counting mode (300 KeV). TOMO software was used to set up data collections, and a dose of -2.4 e$^−$ per image was used (total dose 148 e$^−$/Å$^2$). Projections were motion corrected as above and 3D reconstructions also generated using eTOMO (IMOD). Tomograms were binned by a factor of 4, yielding a final pixel size of 9.6 Å, and representative images ( > 6 tomograms per condition) were Gaussian blur filtered in ImageJ. Segmentation of representative tomograms was performed using AmiraEM software (Thermo Scientific).

## Subtomogram averaging
PEET (IMOD package) was used for STA[49] and Bsoft for basic image processing[50]. Briefly, for both conditions -200 tripodal spikes were initially selected and the spikeInit programme used to calculate initial orientations perpendicular to the viral membrane. These initial orientations were employed to generate an initial 'tripod' reference used for subsequent particle selection.

For the pH 7.3/no K$^+$ tripod STA, the seedSpikes and spikeInit programmes were used to automatically estimate all spike positions and orientations on the viral membranes (71 virions used, -9000 subtomograms), which were then iteratively refined following PEET guidelines using the previously generated initial reference. A summary of the processing stages and EM maps obtained can be found in Supplementary Table 1. Duplicate particles were removed at each stage, and a spherical mask around the tripod was used to focus alignments. After the initial refinements C3 symmetry was evident, therefore in later stages pseudo-particles representing the 3 possible orientations of each tripod were calculated, thereby tripling the number of subtomograms. Additionally, re-creating unbinned tomograms using -30˚ to +30˚ tilt angles, to remove the projections from high tilt angles, improved the resolution achievable in STA. For the whole tripod, this resulted in 20,282 subtomograms with a calculated FSC resolution of -9.1 Å, using the calcFSC PEET programme and a 0.5 cutoff. The subtomogram average of the floor region was generated as above, but by re-centring the average on this region instead of the tripod (using the modifyMotiveList programme to translate the position of the original spikes before tripod refinement) and then iteratively refining the locations and orientations of the subtomograms. This similarly resulted in 16,096 subtomograms, but an improved resolution of -6.6 Å (0.5 FSC cutoff) of this region using the full dataset.

For accurate structural determination and resolution calculations GS-FSC was also performed on half split datasets. For GS-FSC resolution estimation, all automatically selected particles were passed through an initial iteration to remove duplicate particles. The remaining particles were split into half datasets and independently aligned and averaged following identical steps to that outlined above. The half datasets were then aligned to bring the two averages into a common position and orientation, following PEET guidelines. The FSC

between the two half maps was computed using the calcUnbiasedFSC programme (PEET), the result of which was used to filter the resolution of the full dataset. This resulted in an unbiased FSC calculation (using a 0.143 cutoff) for the tripod was ~16 Å and ~13 Å for the floor STA.

For the pH 6.3/K$^+$ treated virus STA, automatic selection of spikes using seedSpikes, did not result in a coherent average. Subtomograms were therefore manually selected (~20,000) from 89 virions and alignment was refined iteratively using the initial reference, obtained as for the control. No symmetry was evident and therefore was not applied (Supplementary Table 1). Additionally, the density at the top of the spikes could not be well refined suggesting multiple conformations. To address this, principal components analysis (pca) and clusterPca programmes were used to identify clusters of separate conformations. Three clusters were identified however the top of the spikes remained unresolved for all (this was also the case for all the clustering options explored) and therefore suggested a highly dynamic structure. The largest cluster (9,987 subtomograms) was taken forward for further refinements utilising a loosely fitted mask to improve the more structured regions and resulted in ~16 Å resolution. This was determined using the calcFSC (0.5 cutoff) PEET programme (not GS-FSC), owing to the high flexibility and low resolution of the average.

Subtomograms were visualised using Chimera and ChimeraX[51,52]. Fitting of the BUNV head domain (pdb: 6H3V), SBV stalk (pdb: 6H3S)[6] and the LACV fusion domains (pre-fusion modelled from pdb: 7A57) was also performed in ChimeraX; using fitmap, which provides cross-correlation scores. Comparison of the RMSD values between the individual domain structures of the BUNV, SBV & LACV Gc with the AlphaFold Gc was obtained using the matchmaker command in ChimeraX. RMSD values (Å) for the pruned atom pairs are shown (Supplementary Fig. 4e).

A local install of AlphaFold_Multimer v2.1.0 (as described at https://github.com/deepmind/alphafold) was used to generate the multimer structure of the BUNV Gn/Gc complex, using the M segment sequences for Gn and Gc from uniprot: P04505 (Gn residues 17-302, Gc residues 478–1433)[24,29]. At the time of model generation, only the Gc head and stalk domains were available for orthobunyaviruses; no Gc fusion domain nor Gn structures were available for training the model. For fitting into the tripod STA, the output model was rotated (ChimeraX and Coot) using rigid-body fitting about the flexible interdomain region (which also had low model confidence) between the stalk and fusion domains I (see Supplementary Fig. 4a, d).

### Reporting summary

Further information on research design is available in the Nature Portfolio Reporting Summary linked to this article.

## Data availability

The data that support this study are available from the corresponding author upon request. The cryo-ET STA averages have been deposited in EMBD under accession codes EMD-15557 (STA of the pH 7.3/no K$^+$ BUNV tripod in Fig. 1); EMD-15569 (STA of the floor region in Fig. 2); and EMD-15579 (STA of the pH 6.3/K$^+$ GPs in Fig. 5). The AlphaFold model (Fig. 3), lacking the TMDs (which are not supported by the STA data) can be found in Supplementary Data 1. The previously published X-ray crystal structures can be obtained from PDB using accession codes 6H3V [https://doi.org/10.2210/pdb6H3V/pdb] (BUNV Gc head domain); 6H3S [https://doi.org/10.2210/pdb6H3S/pdb] (SBV Gc head/stalk domains); and 7A57 [https://doi.org/10.2210/pdb7A57/pdb] (LACV Gc fusion domains in the post-fusion conformation). The source data underlying Fig. 6b, c; and Supplementary Figs. 4a, 6a, 7b–d, 7f, g and 8a–c are provided as a Source Data file. Source data are provided with this paper.

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

## Acknowledgements

Thanks to the Astbury Biostructure Laboratory electron microscopy facility and the bioimaging facility for support in using the microscopes, and a particular thanks to Dr. Emma Hesketh, Dr. Rebecca Thompson, Dr. Yehuda Halfon and Dr. Louie Aspinall (University of Leeds) for the cryo-electron tomography data collection setups. Thanks to Professor Félix Rey (Institut Pasteur, Paris) for kindly providing the structure of the LACV post-fusion Gc prior to publication and for useful comments on the manuscript. Thanks also to Dr Xiaohong Shi, Professor Richard Elliott and Professor Alain Kohl from the Glasgow Centre for Virus Research (CVR, University of Glasgow), and Dr Cheryl Walter (University of Hull) for providing reagents. Finally, thanks to Dr. Samuel Haysom, Dr. Katherine Fenn and Jonathan Machin for their help with AlphaFold. J.F. was funded by the University of Leeds (University Academic Fellow scheme). This work was funded by the Human Frontiers Science Program (RGP0040/2019) awarded to J.F., the Academy of Medical Sciences and the Wellcome Trust (Springboard Award, SBF002\1029) awarded to J.F., the Rosetrees Trust (A1618) awarded to J.F., and the MRC (MR/T016159/1) awarded to J.M., J.N.B. and J.F. Electron Microscopy was performed at the Astbury Biostructure Laboratory (University of Leeds), which was funded by the University of Leeds and the Wellcome Trust (108466/Z/15/Z, 090932/Z/09/Z, 221524/Z/20/Z). The IncuCyte ZOOM live imaging system was funded by the BBSRC (BB/P001459/1), and awarded to Professor Nicola Stonehouse (University of Leeds). The Zeiss LSM 880 upright Confocal microscope was funded by the Wellcome Trust (WT104918MA) and awarded to the University of Leeds BioImaging facility.

## Author contributions

Experimental work performed by S.H.; grid preparation by S.H. and J.F.; liposome preparation by J.J.S.; computational analysis by S.H., J.F., F.W.C., and J.H.; experimental design by S.H., J.F., J.M., and J.N.B.; manuscript preparation by S.H., J.F., J.M., and J.N.B., with input from all authors.

## Competing interests

The authors declare no competing interests.
