## [Peer Review File · Nature Communications]

Organisation of the orthobunyavirus tripodal spike and the structural changes induced by low pH and K⁺ during entryReviewers' Comments:

Reviewer #1:

Remarks to the Author:

The manuscript is focused on the structural characterization of the Bunyamwera virus (BUNV) spikes and its conformational change upon exposure to pH 6.3 with potassium (140 mM), simulating conditions during viral cell entry in endosomes. Using Cryo-ET and subtomogram averaging (STA), the work resolves the spike at 16 Å (improving previously available density maps at 30 Å) and fitted the molecular structure of Gc head and stalk domains (reported previously) with cross-correlation values of 0.7 and 0.82, respectively. Subtomogram averaging (STA) was also performed for the floor region of the spike reaching a 13 Å resolution map. The authors use this map to further fit the modeled Gc fusion domain in its prefusion conformation into the floor region and a AlphaFold-predicted Gn/Gc heterodimer, generating for the first time a very impressive full atomic map of the BUNV spikes. In an additional approach the authors characterize virions treated at pH 6.3 with potassium by Cryo-ET and STA, generating a 16 Å resolution map which shows major conformational changes. The very complete structural study is further broadened by cell infection experiments, with the aim to functionally relate the obtained low-pH/potassium structure with function. However, the used methodological approach by priming wt and GFP-mutant BUNV and then infecting cells with or without previous washing steps seems does not allow for conclusions and should be improved.

Major concerns:

1. Virus aggregation at low pH is a common phenomenon among many enveloped viruses. In figure 2, the authors clearly demonstrate a role of potassium to prevent virus aggregation at pH 6.3 (Figure 2 c,d). Which evidence do the authors have to demonstrate, that potassium has an additional role (e.g. priming/triggering)? Knowing that pH 6.3 without potassium induces viral aggregation, the results of the "cell binding assay" presented in figure 1 are not convincing as such, since the washing step may simply remove the clusters of virus particles. If the authors want to define binding to the cell, then proper cell binding assays should be performed and quantified. Therefore, the GFP-labeled BUNV (Fig. 1D) or SYTO82/DiD-BUNV may be valuable tools. Further, any suggestions on membrane binding should be demonstrated by experiments involving artificial membranes.

2. The functional aspects of the manuscript should distinguish between possible conformational forms of the spikes such as i) priming, ii) intermediate and iii) post-fusion. In previous literature (Ref 21) it has been shown that at pH 5.3 induces glycoprotein-mediated cell-cell fusion and inactivates virus particles irreversibly (Ref 10), presumably by undergoing an irreversible conformational change into its post-fusion form. However, from the data of the manuscript and previous data, it seems less clear, to which functional step a higher pH such as pH 6.3 (with or without potassium) may lead. It is hence fundamental, to characterize the precise fusion activation pH, e.g. by using an acid-bypass assay (Suppl. Figure 1) or other previously described fusion assays. If pH6/K+ would be only a priming step, than it would not be expected to observe fusion and infection in the acid-bypass assay. By contrast, infection may be observed at pH 5.3. (The pH 5.3 should be incorporated as control in all functional assays)

3. The authors should provide more information on the predicted Gn/Gc heterodimer by AlphaFold. The methods section must indicate the used NCBI code and used parameters for prediction (including if they used . In results, the authors should include the predicted aligned error and the manuscript would benefit from indicating the RMSD of the model with the existing X-ray structures for the Gc head and stalk domains. Finally, the cross-correlation values of each fitted model into the STA should be indicated (Figure 5).

Additional concerns:

1. The introduction should be improved by providing more structural information on the BUNV spike structure instead of mentioning them in the results section only (e.g. references 14 and 19).
2. The authors may consider to change the order of the results to improve clarity. They may first describe the pre-fusion form of the spikes including their full-atom model, and then show in vitro

results with the pH 6.3 /potassium form and STA.

3. All results should include the number of biological and/or technical replicates (particularly Fig. 1 , Fig. 2 and Suppl. Fig. 1). Also, the results would benefit from quantification of western blots by densitometry (e.g Fig. 1C, Suppl. Fig 1).

4. It is strongly suggested that the experimental conditions used in Figure 1 (time of infection, potassium concentrations) were previously established and compared with other ions.

5. Considering that the authors generated a huge amount of STA from virions, could they show data on how many spikes are there are on different virions? If 20.000 subtomograms can be taken from 89 virions (line 518), are there over 220 spikes in average on viral particles?

6. From Figure 4 it is unfortunately very difficult to appreciate in the unoccupied densities. Additional views may be incorporated and arrows that highlight the area that could accommodate Gn would be very helpful.

7. In Figures 5-7, it would be supportive to keep the same color scheme shown in Figure 4 for Gc. Gn may be colored in additional color.

8. Figure 5 should highlight the position of the Gn N60 glycan apparently covering the Gc fusion loops.

9. The comparison of Gn/Gc structure with that of and other families from the bunyavirales order and E2 of alphaviruses is highly appreciated (discussion and Figure 6). There is however a small mistake in lines 345-347, since the K⁺ ion is coordinated by Y105 (no E106) in the HNTV Gc structure (not ANDV; the residue is not conserved among hantaviruses).

10. Did the authors attempt to fit the Gn/Gc-fusion-domain heterodimer into the STA obtained at pH6.3/K⁺? Suppl. Figure 7b shows that the density is not entirely occupied by fitting the Gc fusion domain alone.

11. The model of Figure 8 is not sufficiently supported, as long as mayor concern 2 cannot be replied.

Reviewer #2:

Remarks to the Author:

In this work, Hover and colleagues report the effect of high concentrations of potassium (K⁺) and protons (H⁺) on the conformation of glycoproteins from Bunyamwera virus and Hazara virus, which belong to different viral families in the order Bunyavirales. The authors propose that potassium induces conformational changes in the bunyavirus envelope glycoproteins that correspond to an intermediate state, which allows unmasking the fusion unit. The manuscript is not particularly well written, especially the results section which is difficult to follow and would benefit from extensive improvement, not in English but in scientific style. Very often the controls are not mentioned, and it is very difficult to find the information necessary to understand how the experiments were performed and the results obtained. Most data shown are not really novel. Several are computational-based predictions. The only new data are from cryo-electron tomography analyses but the resolutions obtained are only slightly higher than those already published for other bunyaviruses years ago. It is difficult to see what this study adds to what was already known or expected from past results. The discussion is globally overstated. In addition, several points are confusing and raise questions. A non-exhaustive list is given below. While interesting, the manuscript data are too preliminary for Nature Communications.

Specific points

1. The introduction would gain interest if it begins with a paragraph on the biological significance of studying bunyaviruses.

2. The authors use a concentration of 20 mM potassium in their study but what is the physiological concentration of potassium in the endosomal compartments? Is 20 mM physiological at the cellular level? Moreover, at such a high concentration of potassium, what is the ultimate impact on the proton concentration in cells on which the endosomal pH and virus fusion depend? Similarly, 2 hours of pre-

treatment does not make sense. These viruses penetrate their cells in half an hour, three-quarters of an hour maximum.

3. What is the rationale for choosing a pH value of 6.3? This value is the one found in early endosomes while the authors claim that the virus fuses in late endosomes, whose intraluminal pH value is lower than 5.

4. In Figures 1 and S1, controls are collected at 18 hours post-infection while other samples are at 24 hours. Can the authors explain this? In this figure, the controls are very unclear. In addition, there is no quantification and statistic of western blot. No information is provided on the number of repetitions. Which MOI was used? Which cells were infected? Which drug was used? In Figure S1C, the authors claim an increase in nucleoprotein in response to potassium but this is far from evident on the blot shown. In Figure 1C, what does "Ctl" stand for? Does it mean no infection? If so, why does a band appear for the nucleoprotein? Why in Figure 1E, there is almost no difference between the pH \sim 7.3/no potassium and pH \sim 6.3/potassium conditions unlike the results shown in Figure 1C? No statistics are calculated in Figure 1f.

5. Still in Figures 1 and S1, it seems that BUNV can fuse at the plasma membrane in the presence of drugs (whose identity is not specified) following an increase in potassium and pH. It has been recently shown that other orthobunyaviruses can fuse at the plasma membrane (GERV) with acidification without increasing potassium. Can the authors discuss this difference since they seem to consider that their results apply to all bunyaviruses and other viruses in their abstract?

6. In Figure S2D, what do the arrows show for NSm and NSs? From my understanding, this gel shows viral proteins from purified virions, and NSm/NSs are non-structural proteins that are not present in the virions, only expressed in infected cells.

7. In line 163, the authors suggest that potassium prevents the direct interaction of viral glycoproteins with other viruses. It sounds inconsistent with the results shown in Figure 1, which suggest that potassium induces conformational changes critical for virus binding. Could the authors comment?

8. Lines 172-175, it is not obvious from the tomographs shown in Figure 2F that viral glycoproteins are polarized on one side of viral particles. Quantification should be presented to support this statement. Based on these images, it looks rather random.

9. In lines 210-212, it is stated that the post-fusion structure of the La Crosse virus glycoprotein has recently been solved. Conversely, in lines 218-219, the authors suggest that only the head domain of the Gc glycoprotein of the Bunyamwera virus is published. This is confusing.

10. Line 219, why not use the La Crosse virus to interpret the STA map since the partial structure is available?

11. Lines 237-246, this paragraph is extremely hard to follow. It should be reworded. It is puzzling that the authors compare the Gn envelope glycoprotein of orthobunyaviruses to that of all other bunyaviruses which are much larger in size (2-3 times more), i.e., suggesting important differences in their structural organization compared to that of orthobunyaviruses.

12. How were viral particles chosen for the 3D reconstruction since they appeared pleiomorphic under some of the conditions tested?

13. HAZV belongs to the Nairoviridae family, which is a different viral family than Bunyamwera virus and La Crosse virus. Nairoviruses are really different from other bunyaviruses regarding their M polyprotein. Drawing a parallel between orthobunya- and nairoviruses is therefore ambitious, especially since there is no real data shown for HAZV here.

14. Since tomographic approaches can be performed directly in cells, it would be more relevant to image infected cells in the presence or absence of potassium channel inhibitors and focus on late endosomes where viral particles are expected to be located. This would allow seeing the impact of potassium directly in physiological conditions. Using GFP-Gc would allow specific identification of viral particles by fluorescence microscopy prior to electron microscopy analysis.

15. Figure 8 is problematic. The authors postulate that the increase in potassium is necessary for intermediate conformational changes prior to activation by endosomal acidity (pH ~6.3). Except that a value of 6.3 corresponds to a pH in early endosomes where potassium concentration is low (from Figure 8). So, in what order do the conformation changes take place, acidity then potassium or the other way around? This is all confusing.

Reviewer #3:

Remarks to the Author:

In this manuscript Hover et al present a series of experiments characterising the tripodal spike of BUNV. Using priming and infection assays they show that BUNV treated cells (low pH and high K+) lead to increased infection rates possibly due to its ability to escape the endosomal system more readily via increased K+. This provides the authors a way to compare the effects of treated viruses during infection, directly with isolated viruses under similar conditions using high-resolution cryo-ET imaging. They conclude that the glycoproteins undergo a drastic conformational change upon (low pH and high K+) treatment, revealing a fusion intermediate.

The paper is primarily descriptive, but in my opinion nevertheless an important contribution for the following reasons, (1) the study provides the first evidence of where the Gn ectodomain and Gc fusion domain are located within the virion, (2) it demonstrates where the fusion loops are located and insights into how they are being protected by the Gn ectodomain and the Gc tripod. It is also a significant contribution to the understanding of the type 2 fusion proteins in general: observations that the tripod disassembles upon (low pH and high K+) treatment but remains intact. The images/videos are well chosen to illustrate key points and structural features.

The cryo-ET is impressive. The strength of the paper is the elegant use of AlphaFold to predict the entire GP of BUNV. I am in little doubt that this model is correct. This structure has also been observed using cryo-ET before by Bowden et al, albeit at relatively lower resolution, however in this study the authors reveal density for the floor region which they claim represents the Gn ectodomain and Gc fusion domain. The authors support this finding by replacing Gc head and stalk with eGFP and show this virus construct mimicked WT at low pH and high K+. For the sub-tomogram average generated from treated (low pH and high K) viruses, all we are presented with is some weird unidentifiable blob leading to a highly descriptive interpretation of it being a fusion intermediate only supported with results from infection assays using isolated treated viruses as evidence.

The weakness of the paper is the priming and infection experiments (figure 1 and supplementary figure 1b) which distracts the reader from the beautiful cryo-ET and modelling work which in my view should be at the forefront. The work suggests that BUNV penetration into cells depends on K+ activity. There is no question that the unbalanced treatment of a virus can cause abnormalities in virus structure and more importantly trafficking. The authors use an acid bypass assay to show their viruses are not fusing with the plasma membrane, but this is not novel and has been detailed in previous work by the same authors (Hover, 2018). In my opinion it offers no value to the current manuscript and should be removed.

Is there a technical limitation that prevents clear identification of treated viruses inside endosomes? To support their findings and provide more insight the authors could apply cryo-ET to purified

endosome compartments of BUNV infected cells to observe these viruses directly, or at least show that these treated viruses are being trafficked via the endocytic pathway using light-microscopy. I also understand that the authors are perhaps being careful in how they describe the observations, but clearly articulating the caveats and limitations of their infection assays would be beneficial to the field.

One important weakness is that if treated primed viruses exhibit glycoproteins in an intermediate state at the plasma membrane, then what is the cue for fusion once they are taken up in the endosome. By the authors own admission, their previous work shows clearly that K⁺ gradient in the endosome pathway is crucial for the release of viruses into the cytoplasm. The paper suggests some other stimulus is involved in triggering fusion during infection. Despite the high quality of the cryo-ET work, the manuscript is descriptive and without evidence that structural features (fusion and capping loops) observed in their AlphaFold model are indeed correct it remains speculative in most parts. To validate their model and add more evidence to their study I urge the authors to perform more experiments to further support their interpretations.

Specific concerns on more detail.

The title of supplementary figure 1 is pH 6.3/K⁺ treatment of BUNV increases interactions with host cells. However, 1A is the initial test assay just using BUNV at 4 degrees with no treatment.

Do the snowflakes represent 4 degrees?

In Fig1c if the 18 hpi ctl is with no wash (lanes 1 and 2), it might be worth labelling this "no wash" in the figure to make this easier to follow. Same for ctl lane 3.

Line 96. Wouldn't weakly bound be a better word. Unbound suggests a lack of interaction with cells.

Line 59. Shouldn't this be BUNV?

Line 69. Introduction could do with a couple of sentences detailing our current understanding of the fusion mechanism orchestrated by viral fusion type fusion 2 proteins of bunyaviruses. Most of this crucial information is in the discussion.

In the acid-bypass assay the authors add BUNV to cells and unbound virus was removed from media. Is this different from "washing" in supplementary figure 1A where all weakly bound viruses are removed, and no infection is observed? This is not clear in methods.

In supplementary fig 1C at 18 hpi in ctl bypass lanes 1 and 2 why is there now a band in lane 1. It seems that that the authors used the same conditions as in figure 1C but get contrasting results.

The diagram in supplementary figure 1B is not clear and suggests the TCEP is inactivating unbound virions. In the methods the authors mention that TCEP inactivates non-internalized viruses. Are these viruses still bound to the cell surface? Could the authors make this clearer?

Line 103 weakly bound?

Line 152 low magnification or overview micrographs are more suitable.

Line 154 why use this condition (pH 7.3/K⁺). No mention of it in the rest of the manuscript.

In the cryo-EM section of the manuscript the authors could do a better job explaining their statistics, such as how many viruses were imaged.

Line 172. The authors claim that areas of the virion have glycoproteins are excluded. The discussion of

this is purely descriptive presentation of their images.

Supplementary figure 2d markers don't line up with virus protein bands

Was focused classification done on the tripod region. Not clear in the main text.

Regarding using modelling to confirm the hand. What function in chimera X was used to do the modelling? The result of the inverted hand being correct could be the result of incorrect modelling at this resolution.

The authors seem to have gained higher resolution maps using the full data set. Did the resolution drop after splitting data because effectively it's been halved in each average? Would it be worth showing the map of the full data set if it is comparable?

Line 280. Completely missing? Do the authors mean incomplete? What about the Bowden et al manuscript?

Line 282. bunyavirus should be BUNV

Line 306. It is unclear what structure the authors are comparing when discussing length.

RESPONSE TO REVIEWER COMMENTS

Our responses are in blue text. Figures, page and line numbers refer to those from the revised manuscript unless stated otherwise.

Reviewer #1 (Remarks to the Author):

The manuscript is focused on the structural characterization of the Bunyamwera virus (BUNV) spikes and its conformational change upon exposure to pH 6.3 with potassium (140 mM), simulating conditions during viral cell entry in endosomes. Using Cryo-ET and subtomogram averaging (STA), the work resolves the spike at 16 Å (improving previously available density maps at 30 Å) and fitted the molecular structure of Gc head and stalk domains (reported previously) with cross-correlation values of 0.7 and 0.82, respectively. Subtomogram averaging (STA) was also performed for the floor region of the spike reaching a 13 Å resolution map. The authors use this map to further fit the modeled Gc fusion domain in its prefusion conformation into the floor region and a AlphaFold-predicted Gn/Gc heterodimer, generating for the first time a very impressive full atomic map of the BUNV spikes. In an additional approach the authors characterize virions treated at pH 6.3 with potassium by Cryo-ET and STA, generating a 16 Å resolution map which shows major conformational changes. The very complete structural study is further broadened by cell infection experiments, with the aim to functionally relate the obtained low-pH/potassium structure with function.

We thank this reviewer for their positive comments about our subtomogram averaging, BUNV envelope model and infection experiments.

However, the used methodological approach by priming wt and GFP-mutant BUNV and then infecting cells with or without previous washing steps seems does not allow for conclusions and should be improved.

We believe the additional experiments included in the new version of the manuscript address the concerns from this reviewer (see additional experiments under Major concerns 1 and 2 below).

Major concerns:

1. Virus aggregation at low pH is a common phenomenon among many enveloped viruses. In figure 2, the authors clearly demonstrate a role of potassium to prevent virus aggregation at pH 6.3 (Figure 2 c,d). Which evidence do the authors have to demonstrate, that potassium has an additional role (e.g. priming/triggering)?

The effect of K⁺ beyond that elicited by pH 6.3 is clearly demonstrated by the differences (-/+ K⁺) observed in Figures 5a-c, 6b-c and Supplementary Figs 7-8. In addition, our previous papers ((Hover *et al.*, 2016, reference 18 in the revised manuscript; and Hover *et al.*, 2018, reference 19) extensively investigated the role of host-cell K⁺ ion channels and K⁺-treatment of virions in expediting BUNV infection. Within these papers, we determined that: 1) BUNV activates and requires cellular K⁺ channels to infect cells; 2) K⁺ channel function is critical to events shortly after virus entry but prior to viral RNA synthesis/replication; 3) members of the two-pore domain K⁺ channel family play a role in mediating BUNV K⁺ channel dependence; 4) K⁺ channel inhibition alters the distribution of K⁺ across the endosomal system and arrests virus trafficking in endosomes; and that 5) *in vitro* pre-infection treatment of virions with pH 6.3/K⁺ (however not pH 6.3/no K⁺) elicits changes in the virus which expedite endosomal escape.

Knowing that pH 6.3 without potassium induces viral aggregation, the results of the “cell binding assay” presented in figure 1 are not convincing as such, since the washing step may simply remove the clusters of virus particles. If the authors want to define binding to the cell, then proper cell binding assays should be performed and quantified. Therefore, the GFP-labeled BUNV (Fig. 1D) or SYTO82/DiD-BUNV may be valuable tools.

To address the reviewer’s concern regarding the removal of pH 6.3/no K^+ virion clusters by washing steps, we have performed an additional binding assay using immunofluorescence to label surface-bound virions post-washing steps (Supplementary Fig. 8e). In this experiment we show that pH 6.3/no K^+ treated virions remain bound after washing. It is likely that these large clusters are too big to be internalised into an endosome therefore, although they bind, they do not lead to a productive infection and hence the lack of BUNV-N after 24 hrs infection (Supplementary Fig. 8c-d).

Further, any suggestions on membrane binding should be demonstrated by experiments involving artificial membranes.

To further confirm the effects of membrane binding or fusion upon pH 6.3/ K^+ , we additionally performed experiments with artificial membranes, as suggested. As shown in Fig. 6d-f, we combined purified virions with liposomes at different pH and K^+ conditions, and visualised the effects by cryo-ET. Indeed, we clearly identified that pH 6.3/ K^+ induced virus-liposome fusion, as exemplified by the viral GPs and vRNPs now found upon and within liposomes.

2. The functional aspects of the manuscript should distinguish between possible conformational forms of the spikes such as i) priming, ii) intermediate and iii) post-fusion. In previous literature (Ref 21) it has been shown that at pH 5.3 induces glycoprotein-mediated cell-cell fusion and inactivates virus particles irreversibly (Ref 10), presumably by undergoing an irreversible conformational change into its post-fusion form. However, from the data of the manuscript and previous data, it seems less clear, to which functional step a higher pH such as pH 6.3 (with or without potassium) may lead. It is hence fundamental, to characterize the precise fusion activation pH, e.g. by using an acid-bypass assay (Suppl. Figure 1) or other previously described fusion assays. If pH6/ K^+ would be only a priming step, than it would not be expected to observe fusion and infection in the acid-bypass assay. By contrast, infection may be observed at pH 5.3. (The pH 5.3 should be incorporated as control in all functional assays)

To elucidate the conformational step induced by pH 6.3/ K^+ , we performed additional fusion experiments utilising different acidic buffers with a target membrane present; namely the acid-bypass assay and the liposome-virus fusion assay (Fig. 6). As previous work has shown the effects of fusion at pH 5.3, we repeated our previous acid-bypass assay including a pH 5.3/no K^+ and an ‘untreated’ (no acidic buffer) control for both ‘fusion’ and ‘endocytosed + fused’ conditions (Fig. 6a-c). This confirmed that fusion can be induced at the plasma membrane at both pH 6.3 -/+ K^+ ; however + K^+ is the most efficient condition of those we tested. No fusion was observed for the acid-bypass assay at pH 5.3/no K^+ , however the duration of this incubation was only 2 mins. Differences with that seen in the literature (Shi et al., 2007, reference 40) may be a result of differences in time or buffer composition. Of note, we attempted to increase the time of the acid-bypass to 5 mins and/or a pH 5.3 buffer with K^+ , however the cells could not tolerate either condition.

These results were confirmed in the liposome-virus fusion assay, where no receptors or other host factors are present. These assays also suggested that pH 6.3/no K^+ could induce

fusion. Due to the longer incubation time (2 hrs), we did observe fusion at pH 5.3 (Supplementary Fig. 9a), as previously reported. In summary, our results show that fusion can be induced by low pH alone, which is common for many viruses; however, the presence of K^+ in endosomes makes fusion more efficient. Taking into account viruses do not encounter low pH alone during endosome acidification (low pH in endosomes is always accompanied by K^+ in physiological conditions), our findings have direct consequence on our understanding of viral fusion, e.g. the endosomal stage at which viruses fuse might be earlier than what has been described for a range of viruses.

In comparison to these two new experiments, the *in vitro* acid-treatment assays involve treating the virions, and diluting out the buffer before it is introduced to target membranes. We know from our previous work, that after this treatment with pH 6.3/ K^+ , virions still require a reduction in endocytic pH (entry of treated virions can be inhibited by NH_4Cl) and an increase in endocytic K^+ for infection (entry of treated virions can be inhibited by K^+ channel inhibitors), although to a lesser extent (Hover et al., 2018, reference 19). This (and our data from this manuscript) suggests that in the presence of a target membrane (either plasma membrane or a liposomal artificial membrane), pH 6.3/ K^+ can induce fusion; however, when the pH 6.3/ K^+ is applied in the absence of a target membrane, further structural changes after binding to host-cell membranes are required for fusion. We have therefore used this information to generate a new model (Fig. 7) to explain this functional role of pH 6.3/ K^+ and we believe successfully addressed this reviewer's concern.

3. The authors should provide more information on the predicted Gn/Gc heterodimer by AlphaFold. The methods section must indicate the used NCBI code and used parameters for prediction (including if they used . In results, the authors should include the predicted aligned error and the manuscript would benefit from indicating the RMSD of the model with the existing X-ray structures for the Gc head and stalk domains. Finally, the cross-correlation values of each fitted model into the STA should be indicated (Figure 5).

We have updated our description of the AlphaFold prediction as follows:

- Methods: We have added more detail into the methods regarding the generation of the AlphaFold prediction, including the protein sequences and residues used (from UniProt P04505), and how the fitted model was generated (lines 596-603).
- Results: We have also generated a predicted aligned error (PAE) plot (Supplementary Fig. 4c), which defines how confident the prediction is in regards to the location of residue x , in comparison to residue y . In this case AlphaFold is confident within the individual domains (blue), however less confident in-between domains (red), and in particular the position of residues in domains far from one another, for example a residue in the Gc head domain in relation to a residue in the Gc fusion domain. In order to generate this PAE plot, we had to re-run the AlphaFold prediction to include this analysis, as this is not automatically generated on our local installation of AlphaFold; which generated a model with a slightly different angle between the Gc head-stalk and fusion domains than the previous prediction. In addition, we have also calculated RMSD values comparing the head (BUNV), stalk (SBV) and fusion sub-domains I-III (LACV) found in text (lines 151 and 154), which are included as a new table (Supplementary Fig. 4e). The cross-correlation scores are also now indicated in text (lines 144 and 155) and in Supplementary Table 1.

Additional concerns:

1. The introduction should be improved by providing more structural information on the

BUNV spike structure instead of mentioning them in the results section only (e.g. references 14 and 19).

We now describe in the introduction what is known regarding the pre-fusion spike arrangement of *Peribunyaviridae* family members and of other *Bunyavirales* order families (lines 49-67). This information was previously split between the results and discussion sections; we believe this change makes it easier for the reader to understand what is already known about the bunyavirus pre-fusion spike organisation.

2. The authors may consider to change the order of the results to improve clarity. They may first describe the pre-fusion form of the spikes including their full-atom model, and then show in vitro results with the pH 6.3 /postassium form and STA.

We have re-ordered the paper, as suggested by this reviewer. We now focus upon the pre-fusion conformation of the BUNV Gn-Gc glycoproteins and then then move onto the work on the effects of pH 6.3/K⁺. With such a large rearrangement, the movement of areas of text and figures has not been specifically highlighted for clarity; however, below is a summary of the Figure number changes.

Figure Changes Summary

Original Figure	New Figure
1a Endosomal Schematic	Not required
1b-c Binding assay	S8b-c
1d-f GFP-BUNV In vitro treatment	S7e-g
2a-d cryo-EM micrographs	5a-c
2e cryo-ET of pH 7.3 virions	1b
2f-j cryo-ET of virions	5d-f
3a-i STA tripod	1
4 STA floor	2
5 AlphaFold	3
6 AlphaFold comparison with bunyas	4
7 STA pH 6.3/K ⁺	5g-k
8 Model	7 (New Model)
S1a Wash control exp	S8a
S1b-c Acid-bypass	6a-b (b is a new exp)
S2 BUNV purification	S1
S3 Cryo-EM low-magnification	S5
S4a-c Tripod calcFSC	2a-c
S4d pH6.3 K ⁺ calcFSC	S6a
S5 LACV & RVFV Pre- v Post-fusion	S3
S6 AlphaFold pLDDT	S4
S7 pH6.3/K ⁺ STA fitting	S6b-c
S8 Neutralisation assay	S7a-c

3. All results should include the number of biological and/or technical replicates (particularly Fig. 1 , Fig. 2 and Suppl. Fig. 1). Also, the results would benefit from quantification of western blots by densitometry (e.g Fig. 1C, Suppl. Fig 1).

The number of biological repeats is now specified in figure legends and densitometry has been added for the relevant figures (Fig. 6c, Supplementary Figs 7d,g and 8d).

4. It is strongly suggested that the experimental conditions used in Figure 1 (time of infection, potassium concentrations) were previously established and compared with other ions.

As this reviewer has stated, timecourse experiments, K^+ concentrations, comparison with other ions (NaCl and K_2SO_4) were extensively tested in our previously published paper (Hover *et al.*, 2018, reference 19). As suggested by the reviewer, we have made this clearer in the text, both in the results (paragraphs beginning at lines 173 and 231) and the methods sections (line 439).

5. Considering that the authors generated a huge amount of STA from virions, could they show data on how many spikes are there are on different virions? If 20,000 subtomograms can be taken from 89 virions (line 518), are there over 220 spikes in average on viral particles?

The ~21,498 sub-tomograms result from the C3 particle symmetry expansion and correspond to 6,830 tripods from 71 virions. Hence this results in ~96 tripods per virion, matching Bowden *et al.*, 2013, reference 5). To make the whole STA workflow clearer, we have summarised this and different steps of processing in a new Supplementary Table 1.

6. From Figure 4 it is unfortunately very difficult to appreciate in the unoccupied densities. Additional views may be incorporated and arrows that highlight the area that could accommodate Gn would be very helpful.

A new image has been added to show the unoccupied densities more clearly (Fig. 2e) and arrows have been added to indicate the potential location of Gn (Fig. 2e-f).

7. In Figures 5-7, it would be supportive to keep the same color scheme shown in Figure 4 for Gc. Gn may be colored in additional color.

We have simplified the colour scheme to keep it more consistent, while keeping it consistent with the literature. In the literature the colours of the Gc fusion sub-domains (I-III) are consistently red, yellow and blue (e.g. Hellert *et al.*, 2023, reference 26), while the BUNV Gc head and stalk domains are coloured light blue and light green respectively (e.g. Hellert *et al.*, 2019, reference 6). Gn is often coloured in different shades of pink (e.g. Hulswit *et al.*, 2021, reference 10). For whole proteins (Fig. 3) and colouring the maps in Fig. 4g-i colouring each domain would prove too confusing and messy for unfamiliar readers, we therefore decided to use a whole protein colouring scheme for these figures, with Gc as blue and Gn as pink (e.g. Hulswit *et al.*, 2021, reference 10).

8. Figure 5 should highlight the position of the Gn N60 glycan apparently covering the Gc fusion loops.

We have now indicated the position of Gn N60 glycan in Figs. 3a and 3f. As a clarification, N60 is in close proximity to the fusion loop and is not able to fully cover and protect it (line 322-325).

9. The comparison of Gn/Gc structure with that of and other families from the bunyavirales order and E2 of alphaviruses is highly appreciated (discussion and Figure 6). There is however a small mistake in lines 345-347, since the K^+ ion is coordinated by Y105 (no

E106) in the HNTV Gc structure (not ANDV; the residue is not conserved among hantaviruses).

We have now corrected this (lines 368-369).

10. Did the authors attempt to fit the Gn/Gc-fusion-domain heterodimer into the STA obtained at pH6.3/K⁺? Suppl. Figure 7b shows that the density is not entirely occupied by fitting the Gc fusion domain alone.

We have now fitted the Gn/Gc-fusion-domain heterodimer into the pH6.3/K⁺ STA obtained (Supplementary Fig. 6), and included a new table with the cross-correlation scores for each model fitted within the pH 6.3/K⁺ spike (Supplementary Fig. 6e).

11. The model of Figure 8 is not sufficiently supported, as long as mayor concern 2 cannot be replied.

Thanks to the suggestions from all reviewers, and based on the new set of experiments we have included in the revised version of the manuscript, we now propose an updated mechanism for what is happening at pH 6.3/K⁺, both during *in vitro* treatment, and during treatment in the presence of a target membrane. Therefore, the new model in Fig. 7 has been added to reflect this, rather than simply state the general steps of fusion.

Reviewer #2 (Remarks to the Author):

In this work, Hover and colleagues report the effect of high concentrations of potassium (K⁺) and protons (H⁺) on the conformation of glycoproteins from Bunyamwera virus and Hazara virus, which belong to different viral families in the order Bunyavirales. The authors propose that potassium induces conformational changes in the bunyavirus envelope glycoproteins that correspond to an intermediate state, which allows unmasking the fusion unit. The manuscript is not particularly well written, especially the results section which is difficult to follow and would benefit from extensive improvement, not in English but in scientific style. We thank this reviewer for their candor, which we believe has resulted in an improved manuscript. To improve the writing of the manuscript, and in response to comments from the three reviewers, we have re-ordered the results to focus upon the native GP arrangement, followed by the effects of pH 6.3/K⁺ on the BUNV GPs (see additional concern 2 from reviewer 1). As a consequence, we believe that the results section is now much easier to follow and flows more logically.

Very often the controls are not mentioned, and it is very difficult to find the information necessary to understand how the experiments were performed and the results obtained. To make it easier to understand the controls for the biochemical assays (in particular Fig. 6 and Supplementary Figs 7-8), we have improved their description in the figure legends, results section (lines 252 and 270), methods section (line 453, 464, 478 and 483) and on the figures themselves. As with any manuscript, we provided a simplified version of the methods in the figure legends and results sections; however detailed descriptions of how experiments were performed can be found in the methods section.

Most data shown are not really novel.

We disagree with this reviewer about this point. Our manuscript presents a significant amount of novel data. These include: 1) an improved sub-tomogram average of BUNV (from the previously reported ~30 Å to ~16 Å; specifically the floor domain was not specifically resolved in the previous STA; 2) a novel AlphaFold pseudo-atomic model of BUNV envelope, which has been validated in different ways (see comment below); 3) we unambiguously located the Gc fusion domain and its chaperone Gn within the floor domain of the spike (using both, the AlphaFold model and by fitting a model from the related LACV); 4) we show that viral incubation at low pH and K⁺, reminiscent of endocytic conditions, result in a dramatic rearrangement of the BUNV envelope; 5) by employing further structural (STA) and biochemical assays (including acid-bypass, incubation with liposomes and binding assays), we show that while pH 6.3/K⁺ in the absence of a target membrane elicits a fusion-capable 'triggered' intermediate state of BUNV GPs, the same conditions induce fusion when a target membrane is present. Taken together, we reveal new mechanistic understanding of the requirements for bunyavirus entry.

Several are computational-based predictions.

We have used AlphaFold_multimer to generate a model for the BUNV envelope and we believe we have been very clear throughout the manuscript that this is a model. Furthermore, we have validated this model as much as is currently possible, showing the AlphaFold validation and comparison to known structures. In particular, Supplementary Fig. 4 now includes the pLDDT scores for individual residues, a PAE plot (see response to major concern 3 from reviewer 1) and RMSD scores comparing Gc to known structures for

sub-domains from other peribunyaviruses; which are all ~ 1 Å or less. In addition, Fig. 4 compares the individual sub-domains predicted by AlphaFold to those from other *Bunyvirales* families where key similarities have been described in the literature (lines 158-167 and discussed in lines 318-330). With no known structures for full-length Gc, or at all for Gn for peribunyaviruses, we believe this prediction represents a significant advance in the field.

The only new data are from cryo-electron tomography analyses but the resolutions obtained are only slightly higher than those already published for other bunyaviruses years ago. It is difficult to see what this study adds to what was already known or expected from past results.

As mentioned above, we have improved the resolution of the BUNV tripod from ~ 30 Å to ~ 16 Å; and the resolution of the floor domain to ~ 13 Å. This is comparable to the best resolution for a pleomorphic bunyavirus spike (~ 11 Å for Tula hantavirus, Serris et al., 2020, reference 8). Without this newly-derived floor domain average and the AlphaFold model in combination it would not have been possible to identify the peribunyavirus glycoprotein arrangement.

The discussion is globally overstated.

Taking into account the comments from the three reviewers, we have performed additional experiments that have allowed us to propose a new model for the effect of pH 6.3/K⁺ based on the results we have obtained (Fig. 7). We have also rewritten the discussion to avoid any overstatement.

In addition, several points are confusing and raise questions. A non-exhaustive list is given below. While interesting, the manuscript data are too preliminary for Nature Communications.

Below we provide point-by-point responses to each one of the points raised by this reviewer. Combined with the additional experiments we have now incorporated to the manuscript, we now consider our manuscript to be at the level of a Nature Communications article.

Specific points

1. The introduction would gain interest if it begins with a paragraph on the biological significance of studying bunyaviruses.

We have expanded upon our description of the *Bunyvirales* order and the significance of studying this group of viruses (lines 42-46).

2. The authors use a concentration of 20 mM potassium in their study but what is the physiological concentration of potassium in the endosomal compartments? Is 20 mM physiological at the cellular level?

A K⁺ gradient exists across the cell membrane of all cells, with a low (~ 5 mM) extracellular and high (~ 150 mM) intracellular concentration. The exact concentration of K⁺ in specific endosomal compartments is not fully understood, although from the low (~ 5 mM) extracellular concentration, the K⁺ concentration has been shown to increase along the endocytic pathway, with a concentration of ~ 60 mM identified in lysosomes (Scott and Gruenberg, 2011, reference 17). However, we would like to note that traditional methods to determine ion concentration, such as patch clamping, cannot be confidently applied to

determine concentrations within endosomes. Therefore, in this manuscript we opted to employ 140 mM K^+ , which is within physiological range, to mimic the endosomal K^+ concentration. In any case, we have previously shown that treatment of BUNV virions at pH 6.3 with K^+ concentrations ranging between 20 and 140 mM have a similar effect (Hover *et al.*, 2018, reference 19).

Moreover, at such a high concentration of potassium, what is the ultimate impact on the proton concentration in cells on which the endosomal pH and virus fusion depend?

We would like to note that during the *in vitro* pre-infection treatments, the incubation buffers are diluted out prior to addition to cells, as described in the methods (line 445). Therefore, these buffers will not affect the ionic concentrations of the cell. Parts of the results (lines 249-252) and figure legends (Supplementary Figs. 7-8) have been reworded to make this clearer.

Similarly, 2 hours of pre-treatment does not make sense. These viruses penetrate their cells in half an hour, three-quarters of an hour maximum.

The time duration of 2 hrs and the general methodology is based upon previously established methodologies by ourselves for bunyaviruses (Hover *et al.*, 2018, reference 19; and Punch *et al.*, 2018, reference 21) and by others for influenza A virus (Stauffer *et al.*, 2014, reference 34). In addition, although virus penetration (internalisation) into cells occurs quickly, we have previously shown by live-cell imaging of fluorescently-labelled BUNV that virions can be identified within endosomal compartments for longer than 2 hrs (Hover *et al.*, 2018, reference 19). We have reworded part of the results and methods to make it clearer what has previously been established (lines 174-180 and 439-442), including the concentration of 140 mM KCl and the time duration of 2 hrs, both of which are used to maximise the effects observed and minimise any intermediate or heterogeneous structure populations.

3. What is the rationale for choosing a pH value of 6.3? This value is the one found in early endosomes while the authors claim that the virus fuses in late endosomes, whose intraluminal pH value is lower than 5.

The reviewer is correct. BUNV fuses at around pH 6.3/ K^+ , which is the environment of early endosomes (Scott and Gruenberg, 2011, reference 17; and Hover *et al.*, 2018, reference 19). We believe the reviewer is specifically referring to line 195, which was a typo and is now corrected.

4. In Figures 1 and S1, controls are collected at 18 hours post-infection while other samples are at 24 hours. Can the authors explain this?

Fig. 1 and Supp. Fig. 1 are now Supp. Fig. 8 and Fig. 6 respectively. The timepoints are based upon previous work (Hover *et al.*, 2018, reference 19), where 18 hours post-infection (hpi) represents the exponential phase of BUNV viral protein expression and 24 hpi the plateau phase. After pH 6.3/ K^+ *in vitro* pre-infection treatment of BUNV, these virions penetrate cells more quickly and begin producing protein much sooner; therefore for these virions at 18 hpi we observe an increase in BUNV-N compared to the pH 7.3/no K^+ controls. By 24 hpi these levels normalise. In retrospect, we agree that without all the previous data included in the current manuscript this can be hard to follow. We have therefore attempted to summarise this information in the relevant sections of the results (lines 174-180 and 233-237). In addition, for the acid-bypass (Fig. 6b-c) and the binding assay (Supplementary Fig. 8c-d) we have removed the 18 hpi blots for ease in understanding; which were used as an

internal control for us, confirming the successful effects of pH 6.3/K⁺ in every experiment. We have also re-worded the methods section to reflect this and to make it easier to follow (lines 447-450).

In this figure, the controls are very unclear. In addition, there is no quantification and statistic of western blot. No information is provided on the number of repetitions. Which MOI was used? Which cells were infected? Which drug was used?

As stated above (main concern from this reviewer) we have improved the description of the controls in figures and methods sections. Densitometry has also been performed for all western blots and biological repeats included as indicated above (additional concern 3 from reviewer 1). All information requested regarding the MOI and cell lines indicated in the methods (lines 393-394, 445 and 471) and figure legends. Specifically, A549 cells were used for all infection assays. We have also made it clearer the function of the drugs used, which can be found in the figure legend of Fig. 6, results (lines 271-272) and methods sections (lines 485-489). Specifically, we used the reducing agent tris(2-carboxyethyl)phosphine (TCEP; 10 mM, Sigma) to inactivate surface-bound virions and 20 mM ammonium chloride (NH₄Cl), to prevent endosomal acidification and hence entry by endocytosis.

In Figure S1C, the authors claim an increase in nucleoprotein in response to potassium but this is far from evident on the blot shown.

Supplementary Fig. 1c with K⁺ (now Fig. 6b) is the acid bypass assay. The limited increase in N at 18 hpi is likely due to the 2 min incubation, compared to the 2 hrs incubation in the *in vitro* treatment experiments, as stated in the text. As this result was somewhat confusing, as with the previous comment, we have removed the 18 hpi timepoints for ease of understanding. We would also like to note that we have repeated this assay, including more controls (see response to major concern 2 from reviewer 1).

In Figure 1C, what does "Ctl" stand for? Does it mean no infection? If so, why does a band appear for the nucleoprotein?

In Figure 1C (now Supplementary Fig. 8c-d), "Ctl" are virions incubated at pH 7.3/no K⁺. This has now been made clearer in the figure (now labelled 'pH 7'). There is a very faint N band in this condition because most virions are washed in this experiment.

Why in Figure 1E, there is almost no difference between the pH ~7.3/no potassium and pH ~6.3/potassium conditions unlike the results shown in Figure 1C?

Fig. 1e (now Supplementary Fig. 7f) is the *in vitro* pre-treatment while Fig. 1c (now Fig. 6b) is the on-cell acid-bypass assay. As mentioned above, we believe that the difference in incubation time (2 hrs vs. 2 min) and the presence/absence of a target membrane are likely the reason for the two different outcomes. Our manuscript now clearly exemplifies this in Fig. 6 and Supplementary Figs. 7-8.

No statistics are calculated in Figure 1f.

Densitometry and statistical analysis are now shown for all western blots.

5. Still in Figures 1 and S1, it seems that BUNV can fuse at the plasma membrane in the presence of drugs (whose identity is not specified) following an increase in potassium and pH. It has been recently shown that other orthobunyaviruses can fuse at the plasma membrane (GERV) with acidification without increasing potassium. Can the authors discuss

this difference since they seem to consider that their results apply to all bunyaviruses and other viruses in their abstract?

We have repeated our acid-bypass assay (Fig. 6a-c) to include more controls (no-acid/untreated infected control) and acidic buffers (pH 5.3/no K^+), as outlined above (main concern 2 from reviewer 1). This, along with our new liposome-BUNV fusion assay (Fig. 6d-f), now clearly shows that fusion at pH 6.3 is more efficient when K^+ is present, even though fusion can still occur without K^+ . This is in keeping with the Germiston virus (GERV) paper mentioned by reviewer 2 (Windhaber *et al.*, 2022, reference 23), which we have referenced in our manuscript, in which they similarly induced fusion at the plasma membrane (acid-bypass) using pH 6.0 and below. We suggest for BUNV, and potentially other bunyaviruses, that K^+ could make this process more efficient and limit the effects of H^+ to cause stepwise changes in glycoprotein conformations before membrane insertion. It is unfortunate that Windhaber *et al.* did not attempt fusion conditions +/- K^+ as their other results clearly show that “the switch from Na^+ to K^+ within endocytic compartments is important for GERV entry”, which can be prevented using pharmacological inhibitors of host K^+ channels (Windhaber *et al.*, 2022). We have included some relevant information from Windhaber *et al.* in our manuscript (lines 87 and 374-377).

6. In Figure S2D, what do the arrows show for NSm and NSs? From my understanding, this gel shows viral proteins from purified virions, and NSm/NSs are non-structural proteins that are not present in the virions, only expressed in infected cells.

We apologise for this confusion, which we have corrected. The arrows now identify the viral proteins we predict the bands correspond to, rather than the predicted migration distance (Supplementary Fig. 1d).

7. In line 163, the authors suggest that potassium prevents the direct interaction of viral glycoproteins with other viruses. It sounds inconsistent with the results shown in Figure 1, which suggest that potassium induces conformational changes critical for virus binding. Could the authors comment?

(Now line 190) Fig. 5b-c (cryo-EM) demonstrates that virions incubated at pH 6.3/no K^+ cluster together (also shown in Supplementary Fig. 8e), however virions do not cluster at pH 6.3/ K^+ . In the binding assay (Supplementary Fig. 8b-d; old Fig. 1) we are assessing the ability of these viruses to bind to host cells after treatment (including clustering) and then subsequently produce a productive infection. As outlined above (major concern 1 from reviewer 1), although these pH 6.3/no K^+ clusters do still bind to cells (Supplementary Fig. 8e) it is likely that these larger clusters are unable to be internalised into endosomes and hence no productive infection occurs.

8. Lines 172-175, it is not obvious from the tomographs shown in Figure 2F that viral glycoproteins are polarized on one side of viral particles. Quantification should be presented to support this statement. Based on these images, it looks rather random.

We have removed the suggestion that spikes may be polarized.

9. In lines 210-212, it is stated that the post-fusion structure of the La Crosse virus glycoprotein has recently been solved. Conversely, in lines 218-219, the authors suggest that only the head domain of the Gc glycoprotein of the Bunyamwera virus is published. This is confusing.

We agree our previous wording was confusing. We have therefore reworded this to reflect that there are no full length peribunyavirus structures of Gc, and none at all for Gn (lines 136-138).

10. Line 219, why not use the La Crosse virus to interpret the STA map since the partial structure is available?

(Now line 138) We initially did use the LACV structure to interpret our floor domain map, as stated in the text. However, this is a post-fusion structure so initially we had to adjust the orientation of DIII in relation to domains I-II to model a pre-fusion conformation, assuming that this would be the same as for RVFV (Supplementary Fig. 3 and lines 130-133).

Furthermore, we predicted the orientation of the entire fusion domain within the floor region (fusion loops pointing upwards) based upon what is known in the literature for other class II fusion domains.

Most importantly, there are no available structures for a peribunyavirus Gn for us to model within our map and other related viruses possess much larger Gn structures (Fig. 4). We have re-worded the paragraph following this, in line with the above comment, which makes this clearer.

11. Lines 237-246, this paragraph is extremely hard to follow. It should be reworded. It is puzzling that the authors compare the Gn envelope glycoprotein of orthobunyaviruses to that of all other bunyaviruses which are much larger in size (2-3 times more), i.e., suggesting important differences in their structural organization compared to that of orthobunyaviruses.

(Now lines 158-167). We have reworded this paragraph, and moved some of the background information to the introduction. Indeed, the fact that Gn is so much shorter makes it interesting to compare with other bunyavirus families, identifying conserved and different functions between them (also lines 320-332). The individual domain architecture can still be identified when comparing to other bunyavirus families, a fact that has previously been reviewed in Guardado-Calvo and Rey (2021), reference 13.

12. How were viral particles chosen for the 3D reconstruction since they appeared pleiomorphic under some of the conditions tested?

For the tripod/floor spikes were selected using automatic particle picking as the virions are spherical and spikes follow a nearly even distribution across the viral surface. As stated in the methods, “the seedSpikes and SpikeNit programmes were used to automatically estimate all spike positions and orientations on the viral membranes” (lines 552-555). The pH 6.3/K⁺ spikes were chosen by manually selecting spikes from the virion surfaces. This was also stated in the methods: “For the pH 6.3/K⁺ treated virus STA, automatic selection of spikes using seedSpikes, did not result in a coherent average. Subtomograms were therefore manually selected” (lines 578-580). We have now added this information to Supplementary Table 1, along with additional information about image processing.

13. HAZV belongs to the Nairoviridae family, which is a different viral family than Bunyamwera virus and La Crosse virus. Nairoviruses are really different from other bunyaviruses regarding their M polyprotein. Drawing a parallel between orthobunya- and nairoviruses is therefore ambitious, especially since there is no real data shown for HAZV here.

For BUNV, we previously established a role for endosomal K⁺ channels in BUNV entry and established that the endosomal K⁺ concentration is essential for BUNV endosomal escape.

For HAZV we have also determined that endosomal K^+ channels are essential for early stages in HAZV entry and we additionally used cryo-ET and STA to identify structural changes in the HAZV glycoproteins elicited by an elevated K^+ concentration alone (Hover *et al.*, 2016, reference 18; Hover *et al.*, 2018, reference 19; Charlton *et al.*, 2019, reference 20; and Punch *et al.*, 2018, reference 21). Therefore current available data suggests that the increase in endosomal K^+ , a cellular mechanism, is exploited in a similar way by both viruses. Of note, in the current manuscript, we do not say that it is the same, we simply summarise our previous data for BUNV and HAZV (lines 77-85 and 305-307) and offer an interesting comparison (lines 342-344 and 381-384).

14. Since tomographic approaches can be performed directly in cells, it would be more relevant to image infected cells in the presence or absence of potassium channel inhibitors and focus on late endosomes where viral particles are expected to be located. This would allow seeing the impact of potassium directly in physiological conditions. Using GFP-Gc would allow specific identification of viral particles by fluorescence microscopy prior to electron microscopy analysis.

Tomographic approaches can be performed directly upon infected cells, however cryo-FIB milling of whole cells combined with cryo-correlative light and electron microscopy (CLEM) is a time-consuming technique. This suggestion would also be a significant volume of work and data, and while we intend to attempt it in the future, such a project will require a stand-alone paper in its own right. It is therefore beyond the scope of the current manuscript.

15. Figure 8 is problematic. The authors postulate that the increase in potassium is necessary for intermediate conformational changes prior to activation by endosomal acidity (pH ~6.3). Except that a value of 6.3 corresponds to a pH in early endosomes where potassium concentration is low (from Figure 8). So, in what order do the conformation changes take place, acidity then potassium or the other way around? This is all confusing. As stated in response to the major concern 2 from reviewer 1, based on the new set of experiments we have included in the revised version of the manuscript, we now propose a new model in Fig. 7.

Reviewer #3 (Remarks to the Author):

In this manuscript Hover et al present a series of experiments characterising the tripodal spike of BUNV. Using priming and infection assays they show that BUNV treated cells (low pH and high K⁺) lead to increased infection rates possibly due to its ability to escape the endosomal system more readily via increased K⁺. This provides the authors a way to compare the effects of treated viruses during infection, directly with isolated viruses under similar conditions using high-resolution cryo-ET imaging. They conclude that the glycoproteins undergo a drastic conformational change upon (low pH and high K⁺) treatment, revealing a fusion intermediate.

The paper is primarily descriptive, but in my opinion nevertheless an important contribution for the following reasons, (1) the study provides the first evidence of where the Gn ectodomain and Gc fusion domain are located within the virion, (2) it demonstrates where the fusion loops are located and insights into how they are being protected by the Gn ectodomain and the Gc tripod. It is also a significant contribution to the understanding of the type 2 fusion proteins in general: observations that the tripod disassembles upon (low pH and high K⁺) treatment but remains intact. The images/videos are well chosen to illustrate key points and structural features.

The cryo-ET is impressive. The strength of the paper is the elegant use of AlphaFold to predict the entire GP of BUNV. I am in little doubt that this model is correct. This structure has also been observed using cryo-ET before by Bowden et al, albeit at relatively lower resolution, however in this study the authors reveal density for the floor region which they claim represents the Gn ectodomain and Gc fusion domain. The authors support this finding by replacing Gc head and stalk with eGFP and show this virus construct mimicked WT at low pH and high K⁺. For the sub-tomogram average generated from treated (low pH and high K⁺) viruses, all we are presented with is some weird unidentifiable blob leading to a highly descriptive interpretation of it being a fusion intermediate only supported with results from infection assays using isolated treated viruses as evidence.

We thank this reviewer for appreciating our results and for the summary of our results and their impact.

The weakness of the paper is the priming and infection experiments (figure 1 and supplementary figure 1b) which distracts the reader from the beautiful cryo-ET and modelling work which in my view should be at the forefront. The work suggests that BUNV penetration into cells depends on K⁺ activity. There is no question that the unbalanced treatment of a virus can cause abnormalities in virus structure and more importantly trafficking. The authors use an acid bypass assay to show their viruses are not fusing with the plasma membrane, but this is not novel and has been detailed in previous work by the same authors (Hover, 2018). In my opinion it offers no value to the current manuscript and should be removed.

As for reviewer 1, who similarly believes the STA and modelling are the strongest points, we have re-ordered the manuscript to focus on this, followed by the pH 6.3/K⁺ work which has also been extensively improved (additional concern 2 from reviewer 1).

In regards to the acid-bypass assay (Fig. 6a-c), we have repeated this with additional controls to improve the interpretations and conclude that we can induce fusion at pH 6.3/K⁺ at the plasma membrane (main concern 2 from reviewer 1).

Is there a technical limitation that prevents clear identification of treated viruses inside endosomes?

This is an interesting suggestion, however one which would be technically highly-demanding and require a significant body of work to optimise with many variables being out of our control. As such it would occupy an entire paper on its own. As a simpler solution, we have now tested the effect of pH 6.3/K⁺ on virions in the presence of a target membrane, by establishing a liposome-BUNV fusion assay (Fig. 6d-f). This new data clearly showed that these conditions can induce fusion in the presence of a membrane.

To support their findings and provide more insight the authors could apply cryo-ET to purified endosome compartments of BUNV infected cells to observe these viruses directly, or at least show that these treated viruses are being trafficked via the endocytic pathway using light-microscopy.

We have showed previously that treated virions are trafficked via the endocytic pathway. Briefly, NH₄Cl, which inhibits endosomal acidification and hence entry by endocytosis, inhibits entry of *in vitro* treated BUNV at pH 6.3/K⁺, proving that these virions still enter the cell through endocytosis and still require low pH (Hover et al., 2018, reference 19).

I also understand that the authors are perhaps being careful in how they describe the observations, but clearly articulating the caveats and limitations of their infection assays would be beneficial to the field.

We have also re-worded the results sections for the binding and acid-bypass assays to make the interpretations clearer (lines 273-282).

One important weakness is that if treated primed viruses exhibit glycoproteins in an intermediate state at the plasma membrane, then what is the cue for fusion once they are taken up in the endosome. By the authors own admission, their previous work shows clearly that K⁺ gradient in the endosome pathway is crucial for the release of viruses into the cytoplasm. The paper suggests some other stimulus is involved in triggering fusion during infection. Despite the high quality of the cryo-ET work, the manuscript is descriptive and without evidence that structural features (fusion and capping loops) observed in their AlphaFold model are indeed correct it remains speculative in most parts. To validate their model and add more evidence to their study I urge the authors to perform more experiments to further support their interpretations.

As suggested by reviewer 1 (major concerns 1 and 2), we have now performed BUNV-liposome experiments, and additional acid-bypass assays. These additional experiments show that we can induce fusion at pH 6.3/K⁺ however only in the presence of a target membrane at the time of acidic treatment (Fig. 6). The low pH and K⁺ are likely required for subsequent steps after membrane insertion. This is now reflected in our new model (Fig. 7) and the discussion had also been adapted (lines 349-354).

We believe the reference to the lack of structural features observable relates to the effects upon pH 6.3/K⁺, as above this reviewer is happy with the AlphaFold model for the untreated spike and is in "little doubt that the model is correct". Our results and discussion for the role of pH 6.3/K⁺ now focus more on the biochemical assays rather than the STA. The interpretation of which is speculative as we have indicated in text (e.g. in line 214 we say "The lack of density at the top of the spike was indicative of a highly dynamic unstructured region, exhibiting multiple conformations which cannot be resolved by STA"). We have

provided examples for potential structures within this flexible spike (Supplementary Fig. 6) and the only firm conclusions we draw from the STA are the disassembly of the tripod and floor regions (lines 216-220). Beyond this, we interpret our pH 6.3/K⁺ average based upon the biochemical assays.

Specific concerns on more detail.

The title of supplementary figure 1 is pH 6.3/K⁺ treatment of BUNV increases interactions with host cells. However, 1A is the initial test assay just using BUNV at 4 degrees with no treatment.

This figure is now split into Fig. 6 (fusion) and Supplementary Fig. 8 (virus binding). Therefore this comment no longer applies.

Do the snowflakes represent 4 degrees?

They did; but to avoid confusion the schematics have been modified and they now say “4°C” (Fig. 6a and Supplementary Fig. 8b).

In Fig1c if the 18 hpi ctl is with no wash (lanes 1 and 2), it might be worth labelling this “no wash” in the figure to make this easier to follow. Same for ctl lane 3.

As there was significant confusion concerning 18 hpi versus 24 hpi (see specific point 4 from reviewer 2), we have removed the 18 hpi control data for clarity.

Line 96. Wouldn't weakly bound be a better word. Unbound suggests a lack of interaction with cells.

Corrected (line 250).

Line 59. Shouldn't this be BUNV?

Corrected (line 82).

Line 69. Introduction could do with a couple of sentences detailing our current understanding of the fusion mechanism orchestrated by viral fusion type fusion 2 proteins of bunyaviruses. Most of this crucial information is in the discussion.

We agree with this suggestion. Therefore we have moved information regarding the structure of bunyaviruses from other families from the results and discussion to the introduction (lines 49-67) and a short introduction to bunyavirus fusion is found in lines 72-77.

In the acid-bypass assay the authors add BUNV to cells and unbound virus was removed from media. Is this different from “washing” in supplementary figure 1A where all weakly bound viruses are removed, and no infection is observed? This is not clear in methods.

We agree that this was not clear. For this new acid-bypass assay we have re-written the methods which also more accurately reflect how the control wells (endocytosed + fusion) were handled (lines 480-484).

In supplementary fig 1C at 18 hpi in ctl bypass lanes 1 and 2 why is there now a band in lane 1. It seems that that the authors used the same conditions as in figure 1C but get contrasting results.

As mentioned for specific point 4 from reviewer 2, these are not the same conditions and we are comparing the 2 min incubation on-cell acid-bypass (Fig. 6a-c) with a 2 hr *in vitro* pre-infection treatment (Supplementary Fig. 8b-d). Therefore the results likely reflect these differences in incubation time. We have removed the 18 hpi internal control blots as these were also confusing for reviewer 2.

The diagram in supplementary figure 1B is not clear and suggests the TCEP is inactivating unbound virions. In the methods the authors mention that TCEP inactivates non-internalized viruses. Are these viruses still bound to the cell surface? Could the authors make this clearer?

Diagram (Fig. 6a) and methods (line 487) altered to make this clearer. TCEP does inactivate any non-internalised virions, which are those bound to the cell surface as unbound virions are likely removed when adding and removing the acid-bypass buffers.

Line 103 weakly bound?

Wording adjusted (line 482).

Line 152 low magnification or overview micrographs are more suitable.

Wording adjusted to low magnification (lines 184 and 825).

Line 154 why use this condition (pH 7.3/K+). No mention of it in the rest of the manuscript. As this condition is not anywhere in the rest of the manuscript and does not really add anything, we have removed it.

In the cryo-EM section of the manuscript the authors could do a better job explaining their statistics, such as how many viruses were imaged.

To make this clearer, we have added Supplementary Table 1. Specifically, we imaged 71 virions at pH 7.3/no K⁺ and 89 virions at pH 6.3/K⁺.

Line 172. The authors claim that areas of the virion have glycoproteins are excluded. The discussion of this is purely descriptive presentation of their images.

We have removed this statement regarding the polarisation of glycoproteins on the virion envelope.

Supplementary figure 2d markers don't line up with virus protein bands

Corrected (Supplementary Fig. 1d).

Was focused classification done on the tripod region. Not clear in the main text.

No, it was not performed, we re-aligned the centre of the average. We have summarised the processing in Supplementary Table 1 and reworded the methods to reflect this (lines 563-566).

Regarding using modelling to confirm the hand. What function in Chimera X was used to do the modelling? The result of the inverted hand being correct could be the result of incorrect modelling at this resolution.

The 'fitmap' function in ChimeraX was used to fit the head domain trimer into the head domain of the spike (now indicated in methods line 592). We note that the determination of handedness in electron tomography is a known challenge in the field (Briegel et al., 2013,

PMID: 23639902). As described in this paper, the most likely reason for needing to invert the handedness of our average lies within the hardware and software combinations we have employed.

The authors seem to have gained higher resolution maps using the full data set. Did the resolution drop after splitting data because effectively it's been halved in each average? Would it be worth showing the map of the full data set if it is comparable?

As it is standard in the field, we initially processed the whole dataset as one, aligning and averaging all the subtomograms together. To calculate the resolution of this approach, the PEET programme calcFSC splits the dataset in half and compares the two. However, to accurately calculate a resolution (gold-standard procedure, PMID: 22842542) datasets need to be split into two at the start of processing and then independently aligned and averaged, using the same workflow for each half. At the end of this process, the resolution is calculated by comparing the average resulting from each half dataset, and this resolution is used to filter the map resulting from aligning and averaging all particles together (as was done e.g. in Tzviya Zeev-Ben-Mordehai, 2013; PMID: 27035968). For the sake of completion, we have reported the resolutions obtained from both approaches (i.e. gold-standard, which is the resolution mentioned in the text; and the resolution estimated by PEET when using the whole dataset).

Line 280. Completely missing? Do the authors mean incomplete? What about the Bowden et al manuscript?

This is a wording issue, which has now been corrected (line 306). In addition, Bowden *et al* 2013 provide a low resolution map, but no structures.

Line 282. bunyavirus should be BUNV
Corrected (line 309).

Line 306. It is unclear what structure the authors are comparing when discussing length. The length comparisons have now been reworded to make them clearer (lines 63-65). This section now reads: "Compared to these orthologous systems, peribunyaviruses possess a relatively small Gn (~half the size; 186 residue ectodomain for BUNV) and a Gc around twice the size (910 residue ectodomain for BUNV) (Fig. 1a) (9,10)."

Reviewers' Comments:

Reviewer #1:

Remarks to the Author:

The manuscript has now significantly improved and explains in a much more comprehensive way the principal findings. The reviewer further acknowledges the additional functional experiments.

However, several aspects must be improved.

1. The introduction should cite the review of Rey and Guardado Calvo (2021) in which an alphaFold model for Peribunyavirus Gn with missing domain B has already been established. . This should also be included in line 136.
2. The possible functional implications of the conformational change at pH 6.3/K+ is still very much overstated in Lines 301-303 and a statement that their data indicate "fusion loop exposure" must be removed. Although the authors demonstrate "conformational changes" and virus-liposome membrane fusion, it remains to be determined which particular regions of the spikes are involved in the conformational rearrangement of their intermediate structure.
3. Fig. 5. The manuscript should state more clearly in the results section, that in the pH 6.3/K+ STA, "the top of the spikes could not be well refined suggesting multiple conformations". When looking at Fig 5F (top view), there seem to be recognizable "triangles" of shorter distance. Could the authors elaborate more on that part of the structure or quantify the distances of the density?
Also, it would be favorable to better connect this section with the important results of Suppl. Fig.7.
4. Lines 169-199: Please quantify this observation and compare with pH6.3no K+
5. Fig. 5k seems not in "scale" with fig.5E. Please correct
6. Fig. 8e is missing the control pH6.3/no K+ for comparison with Fig. 8c.
7. Lines 280-282/Fig. 6b: Why is there no infection when the virus is allowed to enter by regular endocytosis (lane 6)?
8. Fig. 6F: Please show the segmentation and 3D representation next to the original tomogram.
9. Lines 296-298: Statements on efficiency must be accompanied by quantification or time kinetics.
10. Line 274. Should refer to Fig. 6b. There are several additional incorrect references to figures and supplementary figures throughout the manuscript that must be improved.

Reviewer #3:

Remarks to the Author:

The authors have carefully addressed all my questions and concerns from their original submission. Hover et al. present cryo-ET data showing the architecture of the BUNV Gc-Gn heterodimer within the virus envelope using a complete molecular model using AlphaFold. In the current submission they present new cryo-ET data by combining purified virions with liposomes at different pH and K+ conditions and generated a model to explain the presence of the membrane, low pH and K+ are needed for fusion to occur. The proposed order of events in their model in Fig 7 has been generated from their experiments and interpretation of what they see in the context of existing knowledge. However I find the model in Fig 7 hard to follow and confusing.

- Could the authors use different colors to distinguish the PM from an endosome? Or does the gray membrane only represent PM?
- In a) can the authors state more clearly that their interpretation (presence of the membrane, low pH and K+ at the plasma membrane needed for fusion) is from data generated by the acid bypass assay in the presence of drugs (figure 6, b, lane 4) both in the figure and figure legend?
- In b) can the authors state more clearly that their interpretation is actually from the pH 6.3/no K+ pretreated virus (sup fig 8, c, lane 2). No infection occurs when 6.3/no K+ treated viruses are washed off because they are weakly bound suggesting the fusion loop has not been exposed and the complex needs to be primed with K+ in an extended conformation fusion accessible state to bind strongly to

receptors on the PM. With 6.3/no K⁺ the authors see virus clumping. The authors over interpret their data in the figure legend and make this even more confusing. Their claims in b) that the presence of the membrane, low pH and K⁺ are needed for fusion to occur is not supported by this experiment as it only suggests further processes with 6.3/K⁺ treated viruses in endosomes are needed for fusion and subsequent replication to occur. Panels for b) need to be amended.

- The membrane in the third panel pointed to by the grey dashed arrow (endocytosis?) should be an endosome membrane with a different color?

Other minor comments.

Fig1.i,j are shown on the same scales as the panels in the rest of the figure. The floor region is also difficult to see as the tripod is in the way.

Page 6, line 130. What similarity?

RESPONSE TO REVIEWER COMMENTS

Our responses are in blue text. Figures, page and line numbers refer to those from the revised manuscript without tracked changes.

Reviewer #1 (Remarks to the Author):

The manuscript has now significantly improved and explains in a much more comprehensive way the principal findings. The reviewer further acknowledges the additional functional experiments. However, several aspects must be improved.

1. The introduction should cite the review of Rey and Guardado Calvo (2021) in which an alpha-fold model for Peribunyavirus Gn with missing domain B has already been established. This should also be included in line 136.

This information has been incorporated in the introduction (lines 58-61) and results sections (lines 139-140).

2. The possible functional implications of the conformational change at pH 6.3/K⁺ is still very much overstated in Lines 301-303 and a statement that their data indicate “fusion loop exposure” must be removed. Although the authors demonstrate “conformational changes” and virus-liposome membrane fusion, it remains to be determined which particular regions of the spikes are involved in the conformational rearrangement of their intermediate structure.

Wording has been adjusted to reflect this (lines 314-317, and 404-405).

3. Fig. 5. The manuscript should state more clearly in the results section, that in the pH 6.3/K⁺ STA, “the top of the spikes could not be well refined suggesting multiple conformations”.

Line 220: The wording in relation to Fig. 5 pH6.3/K⁺ STA has been adapted to make this clearer.

When looking at Fig 5F (top view), there seem to be recognizable “triangles” of shorter distance. Could the authors elaborate more on that part of the structure or quantify the distances of the density?

We believe that it is difficult to definitively say if some of these densities are ‘triangles’, however we acknowledge that this is a possibility. What is clear is that the regular arrangement is disrupted and any potentially remaining ‘triangles’ in the floor region are not common and are therefore averaged out during STA processing; even during classification a floor-like class could not be established. We have reflected this observation in lines 225-229.

Also, it would be favorable to better connect this section with the important results of Suppl. Fig.7.

We have reworded lines 250-251 to link the Suppl. Fig. 7 data back to this STA and tomography data (Fig. 5).

4. Lines 169-199: Please quantify this observation and compare with pH6.3no K⁺

The percentage of virions forming clusters has now been quantified (lines 188-189), as well as percentage of virions with GP-free regions of the membrane (line 202).

5. Fig. 5k seems not in “scale” with fig.5E. Please correct

We believe this is in reference to the subtomogram averaging in Figs. 5i, j and k, which we confirm are to the same scale. To avoid confusion, we have added in an additional 2 nm scale bar to Fig. 5i.

6. Fig. 8e is missing the control pH6.3/no K⁺ for comparison with Fig. 8c.

Supplementary Fig. 8e is not a direct comparison experiment to Fig. 8c-d. It is a control to show that the large clusters at pH 6.3/no K⁺ do bind to the cells and are not completely removed

during the washing steps, as was suggested by this reviewer previously.

7. Lines 280-282/Fig. 6b: Why is there no infection when the virus is allowed to enter by regular endocytosis (lane 6)?

There is indeed virus infection in the untreated control where no drugs are present (compare lane 6, to the non-infected 'mock' in lane 1). The intensity of this band is likely limited by the fact that virus is only allowed to enter during the 2 min pulse at 37 °C ('normal' infections are performed for 1 hr at 37 °C, however cells cannot tolerate these buffers for that time), after which non-internalised virus is removed. In addition, the band appears weaker owing to the high intensity of the neighbouring bands in western blot; this intensity allows all bands to be visualised independently and comparably. This information has been reflected in lines 292-295.

8. Fig. 6F: Please show the segmentation and 3D representation next to the original tomogram.

Tomogram has been moved from Supplementary Fig. 9c to Fig. 6g to be presented alongside the segmentation (now Fig. 6h).

9. Lines 296-298: Statements on efficiency must be accompanied by quantification or time kinetics.

Wording has been adjusted to reflect this (line 312).

10. Line 274. Should refer to Fig. 6b. There are several additional incorrect references to figures and supplementary figures throughout the manuscript that must be improved.

The reference in line 284 (previously line 274) has been corrected, as have multiple references to figures throughout the text.

Reviewer #3 (Remarks to the Author):

The authors have carefully addressed all my questions and concerns from their original submission. Hover et al. present cryo-ET data showing the architecture of the BUNV Gc-Gn heterodimer within the virus envelope using a complete molecular model using AlphaFold. In the current submission they present new cryo-ET data by combining purified virions with liposomes at different pH and K⁺ conditions and generated a model to explain the presence of the membrane, low pH and K⁺ are needed for fusion to occur. The proposed order of events in their model in Fig 7 has been generated from their experiments and interpretation of what they see in the context of existing knowledge. However I find the model in Fig 7 hard to follow and confusing.

We have reworded this section (lines 363-373) and the Figure legend to make this clearer.

• Could the authors use different colors to distinguish the PM from an endosome? Or does the gray membrane only represent PM?

In Fig. 7a, the membrane is a target membrane and therefore could represent liposome, cell or endosomal membrane. As such it is coloured grey and is also reflected in the figure legend. In Fig. 7b, this related to the *in vitro* treatment assays and therefore only represents the plasma membrane, we have therefore coloured this blue to highlight the difference. We have also stated this in the figure legend.

• In a) can the authors state more clearly that their interpretation (presence of the membrane, low pH and K⁺ at the plasma membrane needed for fusion) is from data generated by the acid bypass assay in the presence of drugs (figure 6, b, lane 4) both in the figure and figure legend?

This now been specified in lines 363-365 of the discussion and in the figure legend.

- In b) can the authors state more clearly that their interpretation is actually from the pH 6.3/no K⁺ pretreated virus (sup fig 8, c, lane 2).

We have reworded this section (lines 366-373) and figure legend as this is not the case. The interpretation in Fig. 7b is partially as a result of the pH 6.3/K⁺ (not no K⁺) *in vitro* treatment assay in Supplementary Fig. 8b-d, where binding is enhanced and infection is successful after washing steps (this is in comparison to both pH 7.3/no K⁺ lane 1 and pH 6.3/no K⁺ lane 2, neither of which lead to productive infection). This could suggest the virus is able to stably bind to the target membrane, generating a disassembled arrangement as reflected in our cryo-ET and STA results (Fig. 5f-j).

No infection occurs when 6.3/no K⁺ treated viruses are washed off because they are weakly bound suggesting the fusion loop has not been exposed and the complex needs to be primed with K⁺ in an extended conformation fusion accessible state to bind strongly to receptors on the PM. With 6.3/no K⁺ the authors see virus clumping.

The pH 6.3/no K⁺ virions are not washed off, as shown in the new immunofluorescence images in Supplementary Fig. 8e, they are simply not infectious – likely owing to the large size of the clusters at pH 6.3/no K⁺ (Fig. 5b). There are clear differences in terms of clumping between pH 6.3/no K⁺ and pH 6.3/K⁺ (Fig. 5b vs. 5c).

The authors over interpret their data in the figure legend and make this even more confusing. Their claims in b) that the presence of the membrane, low pH and K⁺ are needed for fusion to occur is not supported by this experiment as it only suggests further processes with 6.3/K⁺ treated viruses in endosomes are needed for fusion and subsequent replication to occur. Panels for b) need to be amended.

We have reworded both figure legend and discussion (lines 363-373) to reflect this. Some of the conclusions we have come to are also based upon previously published data in Hover et al., 2018 (ref 19) and as such we have now made this clearer in the discussion. If the low pH and K⁺ are removed before the virus is added to cells (as in Fig. 8b-d), there is no fusion. Low pH and K⁺ are required for subsequent stages in and/or at the time of fusion, as we previously described in Hover et al., 2018 (ref 19). In the liposome assays, the virus, a target membrane (liposome), low pH and K⁺ are the key variables present during these fusion events (Fig. 6e-h). Indeed fusion can occur at low pH alone, however the presence of K⁺ expedites fusion.

- The membrane in the third panel pointed to by the grey dashed arrow (endocytosis?) should be an endosome membrane with a different color?

The arrow represents re-introduction of K⁺ and/or low pH; in our experiments occurring through endocytosis. We have amended the figure and legend to make this clearer.

Other minor comments.

Fig1.i,j are shown on the same scales as the panels in the rest of the figure. The floor region is also difficult to see as the tripod is in the way.

We have rotated Fig. 1e and 1g in order to see the floor domain and unoccupied densities more clearly. We have provided a scale bar for Fig. 1 i and j to demonstrate that they are on separate scales and slightly increased their size. These images are small as they are intended to simply show the C3 symmetry made by three independent tripods, and to shrink the other images to match the size of this would make the details highlighted too hard to be seen.

Page 6, line 130. What similarity?

Wording adjusted on line 132 to specify “amino acid similarity”.

Reviewers' Comments:

Reviewer #3:

Remarks to the Author:

The authors have addressed all of my concerns. The paper is suitable for publication.